# FROM PIXELS TO TOKENS: REVISITING OBJECT HALLUCINATIONS IN LARGE VISION-LANGUAGE MODELS

## ABSTRACT

Hallucinations in large vision-language models (LVLMs) are a significant challenge, *i.e.*, generating objects that are not presented in the visual input, which impairs their reliability. Recent studies often attribute hallucinations to a lack of understanding of visual input, yet ignore a more fundamental issue: the model's inability to effectively extract or decouple visual features. In this paper, we revisit the hallucinations in LVLMs from an architectural perspective, investigating whether the primary cause lies in the visual encoder (feature extraction) or the modal alignment module (feature decoupling). Motivated by our findings on the preliminary investigation, we propose a novel tuning strategy, PATCH, to mitigate hallucinations in LVLMs. This plug-and-play method can be integrated into various LVLMs, utilizing adaptive virtual tokens to extract object features from bounding boxes, thereby addressing hallucinations caused by insufficient decoupling of visual features. PATCH achieves state-of-the-art performance on multiple multi-modal hallucination datasets. We hope this approach provides researchers with deeper insights into the underlying causes of hallucinations in LVLMs, fostering further advancements and innovation in this field.

## 1 INTRODUCTION

Large vision-language models (LVLMs) have demonstrated remarkable performance across a broad range of tasks, even surpassing human capabilities in specific scenarios (Xu et al., 2023; Li et al., 2023a; Zhang et al., 2024a). However, their practical applications are hindered by multi-modal hallucinations, where models generate factually incorrect, inconsistent, or entirely fictitious outputs when interpreting visual features. Recently, various methods have been proposed to address hallucinations in LVLMs, focusing on aspects such as data distribution (Yu et al., 2024; Jiang et al., 2024), training scheme (Zhao et al., 2023; Tong et al., 2024), and decoding strategy (Zhang et al., 2024b; Yang et al., 2024). Despite these advancements, a fundamental question remains unexplored: **What is the primary cause of multi-modal hallucinations?**

To better address the problem, we start by exploring the intrinsic sources of hallucinations in LVLMs. We hypothesize two potential factors: (1) insufficient extraction of visual features and (2) inadequate decoupling of these features during multi-modal integration. To test this, we revisit hallucinations in LVLMs from an architectural perspective, focusing on two key components: the visual encoder (responsible for feature extraction) and the modal alignment module (responsible for feature decoupling). First, we evaluate the role of the visual encoder by combining it with a pre-trained object detection head to perform object recognition on a hallucinatory evaluation dataset. We compare these results to the direct inference results of the LVLM on the same dataset. Our experiments reveal that the primary source of object hallucinations lies in insufficient cross-modal alignment at the projection layer, rather than deficiencies in the encoding capability of the visual encoder. Subsequently, we use the visual encoder to extract object categories and bounding box information. By incorporating the detection information as additional input LVLM sequence, we found that the model's resistance to hallucinations is effectively improved.

Motivated by the above findings, we propose a novel tuning strategy for mitigating hallucinations in LVLMs named **PATCH** (**P**luggable virtu**A**l **T**okens for obje**C**t **H**allucinations). PATCH is designed to help LVLMs more effectively leverage visual detection information, aligning visual and textual features in the semantic space to mitigate object hallucinations. Concretely, PATCH inserts several

trainable and pluggable virtual tokens between image features and enhanced prompt texts, bridging the gap between the encoded image features and input augmented texts with minimal parameter tuning. During inference, the fine-tuned virtual token embeddings are added to the original vocabulary, making PATCH a plug-and-play method that is adaptable to various application scenarios.

To validate the effectiveness and generalization of our method, we conduct experiments on two publicly available multi-modal hallucination evaluation datasets across three mainstream LVLMs. Notably, when enhance the LLaVA-v1.5 (Liu et al., 2024a), MiniGPT-4 (Zhu et al., 2023), and MiniGPT-v2 (Chen et al., 2023a) with PATCH, the accuracy scores on the POPE (Li et al., 2023b) dataset are surged from 85.17% to 90.20% (an absolute improvement of 5.03%), 57.67% to 88.13% (an absolute improvement of 30.46%), and 83.33% to 90.03% (an absolute improvement of 6.70%).

Our experiments and the associated methodology have provided an insightful exploration of the fundamental causes of multi-modal hallucinations from a new perspective, offering new ideas for solving multi-modal hallucinations in LVLMs. In summary, our contributions are three-fold: (1) We explore the intrinsic sources of hallucinations in LVLMs, revealing that the inadequate decoupling of textual and visual features during multi-modal integration is the primary cause of hallucinations in LVLMs. (2) We propose a novel tuning strategy named PATCH, which helps LVLMs more effectively utilize visual detection information to address hallucination problems. (3) The effectiveness of PATCH has been validated on two multi-modal hallucination evaluation datasets across three LVLMs, and the further exploration on various hallucination types has demonstrated its great potential in hallucination mitigation, especially in handling strong misleading difficulty in questions.

## 2  SOURCE ANALYSIS OF MULTI-MODAL OBJECT HALLUCINATIONS

We first identify two potential sources of multi-modal object hallucination in LVLMs. Then, we design and conduct a series of experiments to quantify the impact of these sources. Finally, we thoroughly discuss and analyze the experimental results, offering potential solutions.

### 2.1  POTENTIAL SOURCES OF HALLUCINATION

The architecture of LVLMs typically consists of three components: a visual encoder, a visual projection layer, and a large language model (LLM). The visual encoder is responsible for extracting image features, while the visual projection layer decouples and aligns these features with the semantic space of the LLM. The LLM is responsible for interpreting both the text and image features to generate appropriate responses. Since both the visual encoder and the visual projection layer process image information, they are susceptible to introducing object hallucinations. Therefore, we hypothesize that multi-modal object hallucinations in LVLMs arise from two main sources: (1) **Insufficient extraction of visual features.** The visual encoder may fail to capture critical details or misinterpret objects in the image, leading to inaccurate or incomplete visual representations. (2) **Inadequate decoupling of visual features.** Even when the visual encoder generates accurate features, the visual projection layer may struggle to align these features correctly with the corresponding textual embeddings in the LLM's semantic space. This misalignment hinders the LLM from accurately integrating and interpreting visual information, resulting in hallucinations. To investigate the real source of the hallucinations, we design and conduct the following experiments.

### 2.2  EXPERIMENTS

**Object Hallucination Dataset**  Our experiments are conducted on the POPE (Li et al., 2023b) dataset, which is specifically designed for object hallucination. The dataset consists of 3,000 samples that are highly related to the presence condition of objects. The evaluation of object hallucinations is formulated as a binary classification task, where the model is prompted to respond with "yes" or "no" to questions such as, *"Is there a bicycle in the image?"* With the binary labels, we can directly identify whether the model is hallucinating or not without applying complex parsing rules. If the model answers "yes" for an object not present in the image, or "no" for an object that is present, it is considered to be hallucinating. An example of POPE is illustrated in Appendix A Figure 4.

**Experimental Setup**  We conduct a preliminary experiment by MiniGPT-v2 (Chen et al., 2023a), which employs a vision Transformer (ViT) (Dosovitskiy, 2020) as its visual encoder. To validate the

Table 1: Comparison between number of samples in detection and inference results.

| Results | Correct Inf. | Wrong Inf. |
|---|---|---|
| Correct Detection | 2,396 | 308 |
| Wrong Detection | 105 | 191 |

Table 2: Inference results of $Prompt_1$ and $Prompt_2$ on POPE dataset.

| Prompt | Accuracy | F1 |
|---|---|---|
| $Prompt_1$ | 0.833 | 0.822 |
| $Prompt_2$ | **0.888** | **0.888** |

image encoding capability of MiniGPT-v2's visual module, we combine it with a pre-trained cascade mask R-CNN (Cai & Vasconcelos, 2021) head, which serves as the visual detection model for object recognition. We design a prompt (denoted as $Prompt_1$) as follows to test the zero-shot inference performance of the pre-trained MiniGPT-v2: `[Image]</Img>[vqa][Question]`, where `[Image]` and `[Question]` are placeholders for the input image and question, and `[vqa]` is MiniGPT-v2's task identifier. We then count the number of correct and incorrect samples detected and inferred by the model, as shown in Table 1. Ideally, the detection results should align with the inference results. In other words, correct object detections should correspond to accurate model inferences, while incorrect detections should result in faulty inferences. As a result, any deviation or mismatch between the detected objects and the model's final inference results is identified as an instance of object hallucination.

**Analysis**   As shown in Table 1, 413 samples (308 correct detections with incorrect inferences and 105 incorrect detections) exhibit object hallucinations in MiniGPT-v2, accounting for 13.77% of the total dataset. Among these hallucinatory cases, 74.58% occur when object detection is accurate while the model's inference is incorrect, indicating that this is the predominant failure mode. This suggests that while the visual encoder of MiniGPT-v2 demonstrates a strong capability to extract accurate image features, the visual projection module struggles in seamlessly aligning these features with the LLM. This misalignment between the visual feature encoding and the LLM's interpretation is a key factor contributing to object hallucinations.

**Potential Solutions**   In many computer vision tasks (Hafiz & Bhat, 2020; Deng et al., 2021), objects are typically described by their categories and bounding boxes. We hypothesize that providing these object-related information directly as prompting texts can mitigate the misalignment between visual features and the LLM's semantic space, thus improving the model's understanding of objects in images. To test this hypothesis, we conduct an additional experiment using a modified prompt, denoted as $Prompt_2$: `[Image]</Img>Objects:[Object][vqa][Question]`. In this prompt, object-related information is included at the `[Object]` position, formatted as `category{<x1><x2><x3><x4>}`, where detected object categories are concatenated with their corresponding bounding boxes. This inclusion of object-specific details is intended to enhance the LLM's ability to interpret and process the object information in images. Table 2 shows the results of MiniGPT-v2 using $Prompt_1$ and $Prompt_2$. It is evident that incorporating accurate detection information significantly improves the model's performance on questions about object existence, thus enhancing its image interpretation capabilities and effectively reducing object hallucinations.

**Motivation**   While our preliminary experiments identify the potential causes of object hallucinations and suggest a possible solution, we observe that directly incorporating detection information into the input may introduce unnecessary information, particularly when the detection objects are weakly correlated with the given question. This redundancy may interfere with the model's reasoning process, hindering its ability to effectively utilize the provided information. To address this issue, we propose the PATCH tuning strategy, which employs trainable and pluggable virtual tokens to help the LLM filter and optimize the use of detection information. These virtual tokens serve as a bridge, which enables the LVLMs to focus on task-relevant image features, improving alignment between visual and textual representations within the semantic space.

## 3   METHODOLOGY

In this section, we will first introduce the formulation of the task and then describe specific details of our proposed method, including its underlying principles and implementation steps.

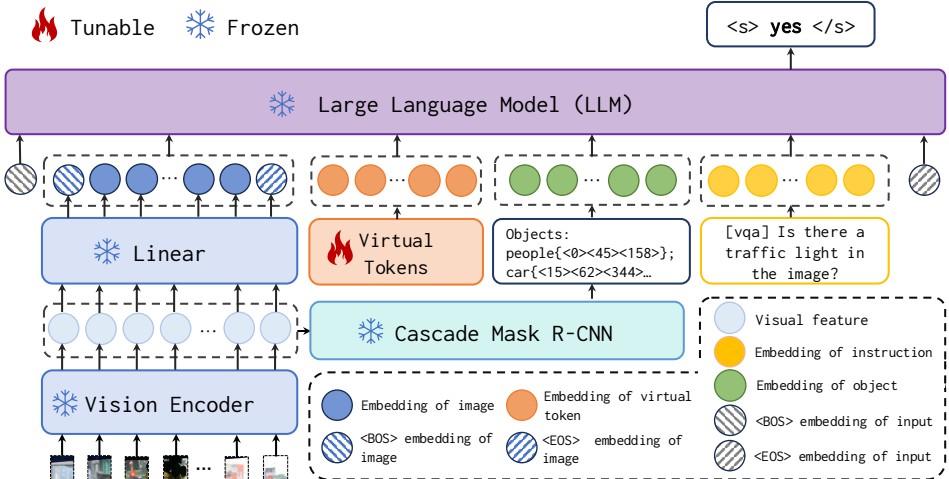

Figure 1: The architecture of LVLMs with PATCH (taking MiniGPT-v2 as an example) where the visual encoder, linear projection layer, and the LLM remain frozen during the training phase. The only updated component during fine-tuning is the parameters of the virtual tokens. A frozen pre-trained Cascade Mask R-CNN head is adopted to obtain the object information in the test images.

## 3.1 PROBLEM FORMULATION

The large vision-language models (LVLMs) aim to generate proper text responses to multi-modal inputs, typically combining visual and textual data. The standard approach involves extracting visual features through a visual encoder, mapping these features into the text semantic space by a visual projecting layer, and performing cross-modal fusion and alignment of both modalities in the semantic space. The fused representations are then decoded by an autoregressive language model to generate the final response. Formally, given an input image $I$ and a corresponding question or text prompt $Q$, the generated answer sequence $Y$ is calculated as:

$$p(Y) = \prod_{t=1}^{T} p_\theta(y_t \mid I, Q, y_{<t}), \tag{1}$$

where $y_{<t}$ denotes the sequence of tokens prior to the current token $y_t$ at step $t$, and $\theta$ represents the parameters of the LVLM. Based on our preliminary analysis, we observe that incorporating object categories $C$ and their corresponding bounding boxes $B$ from the image's object detection results can improve the generation process. The generation process can be reformulated as:

$$p(Y) = \prod_{t=1}^{T} p_\theta(y_t \mid I, D, Q, y_{<t}), \tag{2}$$

where $D = [C(I); B(I)]$. In the task of object hallucination detection, the input usually consists of questions $Q$ designed to assess the presence of hallucinated objects. The goal is to generate the correct answer $Y$ by evaluating whether a given object is present in the image $I$.

## 3.2 PATCH

The proposed PATCH strategy aims to mitigate object hallucinations in LVLMs by introducing trainable virtual tokens that leverage additional object detection information. Specifically, the architecture of our method (taking MiniGPT-v2 as an example) is shown in Figure 6. Inspired by Zhu et al. (2024), we insert a set of $n$ virtual tokens $T = [t_1, t_2, \ldots, t_n]$ between the image features $V$ and the detection information $D$. The embeddings of these tokens are optimized during training, with parameters $\delta \in \mathcal{R}^{n \times d}$, where $d$ is the token embedding size of the LVLM. The generation process of LVLMs, augmented by these virtual tokens, is formulated as:

$$p(Y) = \prod_{t=1}^{T} p_{\delta,\theta}(y_t \mid I, [t_1, t_2, \ldots, t_n], D, Q, y_{<t}). \tag{3}$$

To reduce the computing resources, all parameters $\theta$ of LVLM are frozen during training, except for the newly introduced parameters $\delta$ of virtual tokens. For instance, with the addition of 20 virtual tokens, only $20 \times 4,096 = 0.08$M parameters are trainable, accounting for just 0.0012% of the total model parameters. This significantly reduces the computational costs while preserving the notable optimization effects on multi-modal object hallucinations, details are demonstrated in Section 4.3.

In the inference phase, we extend the model's vocabulary by incorporating several special tokens (*e.g.*, [ref1], [ref2],..., [refn]) whose embeddings are initialized by the fine-tuned virtual token embeddings. This makes PATCH a plug-and-play method that can be dynamically adjusted based on the requirements of applications. Specifically, when detection information is equipped in the users' input, virtual tokens can be added before the detection results, effectively helping to mitigate object hallucinations in LVLMs. In scenarios where no extra detection information is required, the LVLM can revert to processing the input using its standard capabilities without PATCH involvement. This flexibility is especially valuable in practical applications, as LVLMs are commonly deployed for various downstream tasks. Our approach strengthens these models' image comprehension abilities without disrupting their core features or inherent capabilities.

The PATCH strategy enhances the model's ability to utilize detection results, enabling LVLMs to interpret image content with greater accuracy. By narrowing the representational gap between visual features and text in the semantic space, PATCH optimally aligns cross-modal features, particularly for tasks that benefit from enriched detection prompts. This improved alignment strengthens the model's overall performance in reducing hallucinations.

## 4 EXPERIMENTS

### 4.1 DATASETS AND EXPERIMENTAL SETTINGS

**Datasets**   To verify the effectiveness of our method, we conduct experiments on two publicly available multi-modal hallucination evaluation datasets. (1) The **POPE** dataset (Li et al., 2023b) is specifically designed for evaluating object hallucinations in LVLMs. It provides 3,000 samples constructed by adversarial setting for the MSCOCO and A-OKVQA datasets, respectively. We treat the former as the training set while the latter for testing. (2) The **PhD** dataset (Liu et al., 2024c) is a newly introduced benchmark for evaluating multi-modal hallucinations in LVLMs. We conduct experiments on its v1 version across five task types that are highly related to objects, including Object Recognition, Attribute Recognition, Counting, Positional Reasoning, and Sentiment Analysis. We randomly select 80% of the data for training while the rest for testing. Further details can be found in Appendix A.

**Implementation Details**   LLaVA-v1.5, MiniGPT-4, and MiniGPT-v2 are adopted as our backbone LVLMs, initialized by their default parameter configurations. Given the advanced capabilities of MiniGPT-v2, most of our experiments are conducted by this model. We add 20 virtual tokens by default during the fine-tuning and inference phase, and the impact of the token number will be discussed in Section 4.4. Accuracy, Precision, Recall, and F1 score are employed as evaluation metrics. Further details about the experimental settings can be found in Appendix B.

### 4.2 BASELINES

We consider seven mainstream LVLMs: (1) **mPLUG-Owl** (Ye et al., 2023) is a two-stage training framework that aligns visual and textual modalities by training a visual knowledge module followed by an abstraction module. (2) **Multi-modal-GPT** (Gong et al., 2023) utilizes a specially designed input sequence and calculates the loss based on the response and the end token to enhance visual-text alignment. (3) **InstructBLIP** (Dai et al., 2023) employs a Q-Former (Li et al., 2023a) to compress visual features into a fixed number of tokens, which are concatenated with text tokens. This combination allows for the comprehension of visual and linguistic information via instruction-tuning. (4) **LLaVA** (Liu et al., 2024b) and (5) **Minigpt-4** (Zhu et al., 2023) use a simple projection layer for cross-modal embedding alignment. LLaVA emphasizes fine-tuning for visual-language alignment by specific instructional datasets, while MiniGPT-4 is optimized for generating detailed descriptions from images. (6) **LLaVA-v1.5** (Liu et al., 2024a) and (7) **Minigpt-v2** (Chen et al., 2023a) are the improved version of the origin model, with optimizations for better cross-modal understanding.

Table 3: Performance of LVLMs and hallucination solving methods on the POPE dataset. The best results are in **bold**. The values in the subscript indicate the improvement over the backbone model.

| Model | Accuracy | Precision | Recall | F1 |
|---|---|---|---|---|
| mPLUG-Owl | 50.67 | 50.34 | 99.33 | 66.82 |
| Multi-modal-GPT | 50.00 | 50.00 | **100.00** | 66.67 |
| InstructBLIP | 74.37 | 67.67 | 93.33 | 78.45 |
| LLaVA | 50.77 | 50.39 | 99.87 | 66.98 |
| LLaVA-v1.5 | 85.17 | 89.93 | 79.20 | 84.23 |
| + HA-DPO | $81.46_{[-3.71]}$ | $77.99_{[-11.94]}$ | $87.66_{[+8.46]}$ | $82.54_{[-1.69]}$ |
| + HACL | $86.54_{[+1.37]}$ | $\mathbf{93.01}_{[+3.08]}$ | $79.52_{[+0.32]}$ | $85.73_{[+1.50]}$ |
| + Hard Prompt | $89.93_{[+4.76]}$ | $91.14_{[+1.21]}$ | $88.47_{[+9.27]}$ | $89.78_{[+5.55]}$ |
| + PATCH (ours) | $\mathbf{90.20}_{[+5.03]}$ | $91.13_{[+1.20]}$ | $89.07_{[+9.87]}$ | $\mathbf{90.09}_{[+5.86]}$ |
| MiniGPT-4 | 57.67 | 54.29 | 96.93 | 69.60 |
| + HA-DPO | $75.66_{[+17.99]}$ | $74.36_{[+20.07]}$ | $78.33_{[-18.60]}$ | $76.29_{[+6.69]}$ |
| + Woodpecker | $82.33_{[+24.66]}$ | $83.92_{[+29.63]}$ | $80.00_{[-16.93]}$ | $81.91_{[+12.31]}$ |
| + HACL | $71.32_{[+13.65]}$ | $70.53_{[+16.24]}$ | $73.45_{[-23.48]}$ | $71.96_{[+2.36]}$ |
| + Hard Prompt | $70.73_{[+13.06]}$ | $63.60_{[+9.31]}$ | $96.93_{[+0.00]}$ | $76.81_{[+7.21]}$ |
| + PATCH (ours) | $\mathbf{88.13}_{[+30.46]}$ | $\mathbf{86.99}_{[+32.70]}$ | $89.67_{[-7.26]}$ | $\mathbf{88.31}_{[+18.71]}$ |
| MiniGPT-v2 | 83.33 | 88.28 | 76.87 | 82.18 |
| + Hard Prompt | $88.77_{[+5.44]}$ | $88.23_{[-0.05]}$ | $89.47_{[+12.60]}$ | $88.84_{[+6.66]}$ |
| + PATCH (ours) | $\mathbf{90.03}_{[+6.70]}$ | $\mathbf{91.39}_{[+3.11]}$ | $88.40_{[+11.53]}$ | $\mathbf{89.87}_{[+7.69]}$ |

In addition to these LVLMs, we compare our method with three recently released hallucination solving methods based on LLaVA-v1.5 and MiniGPT-4: (1) **HA-DPO** (Zhao et al., 2023) reformulates the hallucination problem as a preference selection task, where the model is trained to consistently favor the accurate response over the hallucinatory one with the given image-question pair. (2) **Woodpecker** (Yin et al., 2023) proposes a five-stage method to locate the hallucinations and determine the facts, introducing a training-free method to correct hallucinations from the generated texts. (3) **HACL** (Jiang et al., 2024) integrates contrastive learning into the training process by using hallucinatory text as hard negative examples. This trains the model to bring the representations of non-hallucinatory text closer to their corresponding images.

As demonstrated in our preliminary experiments (Section 2.2), incorporating object-related information directly into the input prompt (Prompt$_2$) can also lead to performance improvements. We refer to this simple approach as the "**Hard Prompt**" method.

## 4.3 EXPERIMENTAL RESULTS

**Analysis on POPE** The results are shown in Table 3. It is clear to see that our proposed PATCH method significantly improves the performance of all three backbone LVLMs on the object hallucination detection task. From the results, we have the following observations: (1) Pre-trained LVLMs, such as mPLUG-Owl, Multi-modal-GPT, InstructBLIP, MiniGPT-4, and LLaVA are prone to generate hallucinated content, while their advanced versions (LLaVA-v1.5 and MiniGPT-v2) can perform better. We attribute this to their more advanced training strategies (such as instruction tuning). (2) Both the Hard Prompt method and our PATCH method improve the performance of the backbone LVLMs, confirming that incorporating object-related information aids LVLMs in better interpreting visual features. The proposed PATCH can outperform Hard Prompt, as its soft prompt is optimized during the generation task. This enables more precise alignment between image content and the corresponding text in the semantic space, significantly mitigating object hallucinations in LVLMs. (3) Compared to three previous approaches designed to alleviate object hallucinations, our PATCH method achieves a remarkable improvement of 30.46%, 5.03%, and 6.70% in accuracy on LLaVA-v1.5, MiniGPT-4 and Minigpt-v2, respectively, achieving the state-of-the-art performance. This highlights the strong generalizability of our approach. Unlike HA-DPO and HACL, which rely on complex optimization techniques, PATCH achieves performance improvements with minimal parameter tuning. While Woodpecker offers a training-free solution, its reliance on manually crafted hard prompts limits its flexibility and scalability. In contrast, PATCH introduces a soft prompt tun-

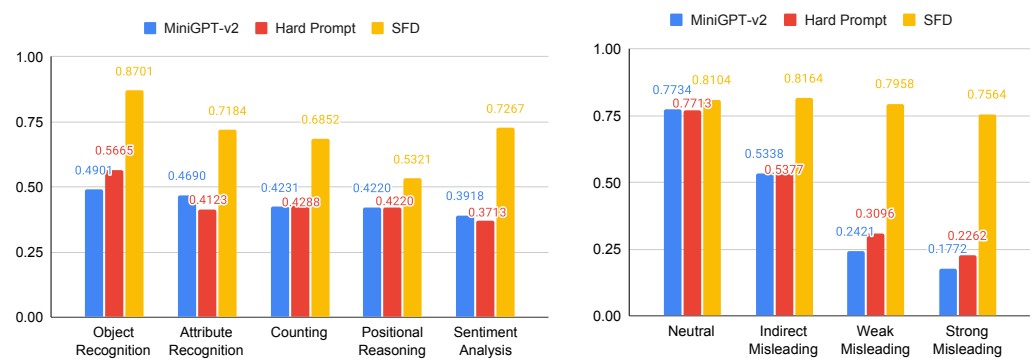

Figure 2: Performance on the PhD dataset across different task types (left) and conflict levels (right).

ing strategy via pluggable virtual tokens, allowing the model to adaptively learn and extract valuable information from detection data during fine-tuning process.

**Analysis on PhD** We compare our proposed PATCH with the Hard Prompt method based on MiniGPT-v2 across different task types and conflict levels on the PhD dataset: (1) **Task types**. The accuracy scores of the three variants across five task types are presented on the left side of Figure 2. We can see that using a hard prompt to inject object-related information can help in object recognition, counting, and positional reasoning tasks. This is consistent with our findings in preliminary experiments, where object-related information can improve the LVLM's understanding of geometric information of objects in images. However, for object attribute recognition and sentiment analysis tasks, the hard prompt brings some negative effects. This may be due to the fact that the detection results do not include information related to the questions. Directly adding the additional information into the input texts may introduce excessive redundant noise, interfering with the ability of LVLMs to leverage prior knowledge for the questions. Fortunately, our proposed PATCH significantly improves performance across all tasks. This clearly demonstrates that the fine-tuned virtual tokens can effectively help the model utilize the valuable information among additional detection results as well as mine the prior knowledge of the LLM. (2) **Conflict levels**. On the right side of Figure 2, we present the accuracy scores across four conflict levels of the three variants. The PhD dataset provides three statements that are completely inconsistent with the image content for each question, and these statements are used as context information to mislead the model during answer generation. The tones of these statements are divided into three types: strong misleading, weak misleading, and indirect misleading, while neutral represents the original question text without added misleading information (An example is shown on the right side of Figure 5 in Appendix A). From the results, it is obvious that the performance of the three variants is relatively close on neutral questions, indicating that recently pre-trained LVLMs are able to answer questions accurately in the absence of misleading information. However, when more misleading statements are added as the contexts, the performance of MiniGPT-v2 and Hard Prompt drops significantly. This shows that LVLMs are struggling in handling questions with misleading texts, and simply adding detection information as prompts are insufficient to address the influence of misleading statements. In contrast, our proposed PATCH method performs remarkably well even on strongly misleading questions, showing that fine-tuning LVLMs with trainable virtual tokens can effectively improve the model's ability to discern the true relationships between textual and visual contents. This demonstrates the robust cross-modal information understanding capability of PATCH, showcasing its effectiveness in addressing the object hallucination issue in LVLMs.

## 4.4 FURTHER ANALYSIS

In this section, we further explore the impact of different virtual token configurations on LVLMs. All experiments are conducted based on the MiniGPT-v2-7b model.

**Impact of Different Object-related Information** We explore the impact of various object detection results on our PATCH method by removing them from the prompts. The results are shown in Table 4. We use "bboxes" to denote the bounding box information and "categories" to represent

Table 4: The ablation results on detection information and virtual token positions on POPE dataset.

| Method | Accuracy | Precision | Recall | F1 |
|---|---|---|---|---|
| PATCH | **90.03** | **91.39** | 88.40 | **89.87** |
| *w/o* bboxes | 88.27 | 87.37 | 89.47 | 88.41 |
| *w/o* bboxes & categories | 82.60 | 84.93 | 79.27 | 82.00 |
| *w/* Late | 87.60 | 85.79 | **90.13** | 87.91 |

the object category information. The results show that when complete detection information is provided, PATCH achieves the highest accuracy of 90.03% on the POPE dataset. This demonstrates that our method effectively enhances the LVLMs' understanding of scenes and objects. When the bounding box information is removed, the accuracy drops by 1.76%, suggesting that bounding boxes play an important role in object localization and scene comprehension. However, a slight increase in the recall score is observed at the same time, suggesting that without bounding boxes, the model becomes more effective at detecting positive samples. This may be because bounding boxes can sometimes cause the model to focus too narrowly on specific regions, limiting its understanding of the broader image context. When all object-related information is omitted, leaving only 20 trainable virtual tokens for fine-tuning, the model's performance drops significantly. This indicates that without the additional prompt texts, the fine-tuned virtual tokens alone are insufficient to achieve robust and consistent semantic alignment between visual and textual features. As a result, the model faces increased uncertainty in object recognition, underscoring the importance of incorporating rich object-related information to mitigate hallucinations in LVLMs.

**Impact of Token Position**  In our PATCH method, virtual tokens are added between the detection results and the question, following the format: "`[Image][Virtual Tokens][Object][Question]`". To investigate the effect of virtual token positions, we consider an alternative format: "`[Image][Object][Virtual Tokens][Question]`", referred to as "PATCH *w/* Late". The results of this comparison are shown in Table 4, which clearly indicates that the position of the virtual tokens significantly impacts model performance. Placing the token between the detection results and the image features, as done in PATCH, allows the LVLM to better interpret and integrate the detection information. In contrast, when placing the virtual tokens after the detection results, the LVLM tends to focus more on the detection texts, while overlooking critical contextual cues from the overall image features, resulting in a performance decline.

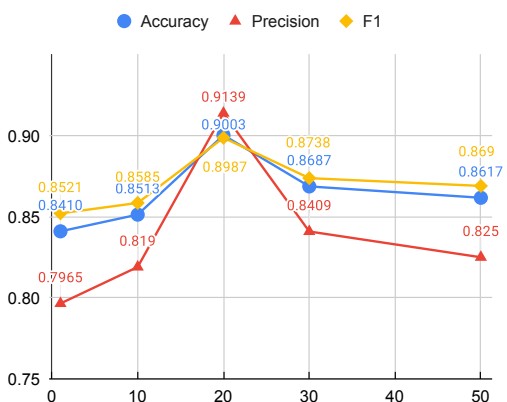

Figure 3: Accuracy, Precision, and F1 score over different token quantities on POPE dataset.

**Impact of Token Initialization**  In our experiments, we find that the initialization texts of virtual tokens may influence the final performance. To investigate this effect, we evaluate PATCH with three different initialization strategies, as shown in Table 5. The results indicate that the random initialization method leads to a noticeable decline in overall model performance. In contrast, when the tokens are initialized with explicit, text-based prompts, all evaluation metrics are consistently improved. Given that the responses in the POPE dataset are constrained to "yes" or "no", we specifically emphasize the initial template as $T_2$. The results in the third row of the table demonstrate that initializing virtual tokens with prompt texts aligned to the answer distribution enables the LVLM to better facilitate virtual tokens, leading to more reliable and standardized response generation.

**Impact of Token Quantity**  We further evaluate the performance of PATCH across different quantities of virtual tokens. As illustrated in Figure 3, the metric scores initially improve as the number of virtual tokens increases, reaching optimal accuracy with 20 tokens, which represents a 6.70%

Table 5: The results of different initialization methods for virtual tokens on POPE dataset.

| Initialization | Accuracy | Precision | Recall | F1 |
|---|---|---|---|---|
| Random | 86.77 | 84.28 | 90.40 | 87.23 |
| $T_1$: According to the previous object detection results, please answer the following question: | 88.83 | **93.76** | 83.20 | 88.17 |
| $T_2$: According to the previous object detection results, please answer the following question with 'yes' or 'no': | **90.03** | 91.39 | **88.40** | **89.87** |

improvement over the baseline configuration. However, further increases in token quantity lead to a noticeable decline in performance. This suggests that an excessive number of tokens may introduce redundant information, which hampers the model's capacity to accurately focus on essential information, ultimately leading to a significant degradation in overall performance.

## 5 RELATED WORK

**Large Vision-Language Models** LVLMs represent a significant advancement in the field of artificial intelligence, combining visual perception and natural language understanding to process and generate contextually rich information. For the past few years, LVLMs have demonstrated remarkable success in a variety of tasks, including image captioning (Chen et al., 2023b), visual question answering (Li et al., 2024), and visual reasoning and generation (Chen et al., 2024). A critical aspect of LVLMs is their cross-modal alignment mechanism, which is the key component that ensures seamless integration of visual and textual embeddings. Models like InstructBLIP (Dai et al., 2023) and LLaVA (Liu et al., 2024b) emphasize fine-tuning based on specific instruction texts or tasks such as detailed image-text generation, helping mitigate the hallucination problems by enhancing the alignment between visual and textual representations. In this paper, we have conducted several preliminary investigations to explore the root mechanisms that drive hallucinations, proposing a new fine-tuning strategy to alleviate the object hallucination issue in LVLMs.

**Object Hallucination in LVLMs** Recently, growing attention has been paid to the hallucination phenomenon which is the direct issue that affects the reliability of LVLMs. Jiang et al. (2024) have analyzed the representation distribution in LVLMs for both text and visual tokens, identifying that there exists a misalignment in cross-modal representations. Zhou et al. (2023) have analyzed the hallucinated textual outputs of LVLMs, finding that the object hallucination issue occurs closely tied to the inherent uncertainties during the beam search process, suggesting that both data distribution and decoding methods contribute to the issue. Most of the previous works focus on optimizing the data distribution (Yu et al., 2024) (Jiang et al., 2024), the training scheme (Zhao et al., 2023) (Huang et al., 2024) or the decoding strategy (Zhang et al., 2024b) (Yang et al., 2024). However, these approaches often require large training resources or artificially defined hard prompts for model tuning. In this paper, we propose a novel method PATCH, which is a minimal parameter tuning strategy, extendable in overcoming the multi-modal hallucination in a variety of application scenarios.

## 6 CONCLUSION

In this paper, we revisited hallucinations in LVLMs from an architectural perspective, proposing two potential causes: 1) insufficient extraction of visual features and 2) inadequate decoupling of visual features. Based on our preliminary experiment results, we identified that the primary cause lies in insufficient cross-modal alignment rather than deficiencies in the visual encoding process. Motivated by this insight, we introduced the PATCH tuning strategy, which leverages trainable virtual tokens to effectively bridge the semantic gap between the encoded image features, detection information augmented prompts and questions with minimal parameter tuning. By incorporating the fine-tuned virtual tokens into the LVLM vocabulary, PATCH became a versatile plug-and-play method that can be easily applied across various tasks. Extensive experimental results on two publicly available datasets demonstrated the effectiveness and generalization of our approach in handling different levels of misleading difficulty in questions and addressing the object hallucination issue in LVLMs.

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

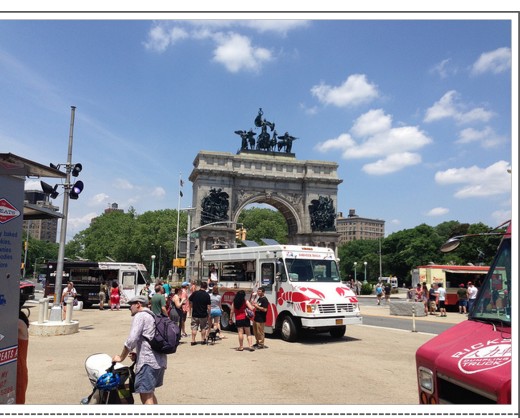

Figure 4: The example of question-answer pairs sampled by adversarial setting in the POPE dataset. For each image, the dataset proposes six questions, three of which are positive samples. The corresponding negative samples are produced by replacing the object in each positive sample with a not present frequently co-occurring object.

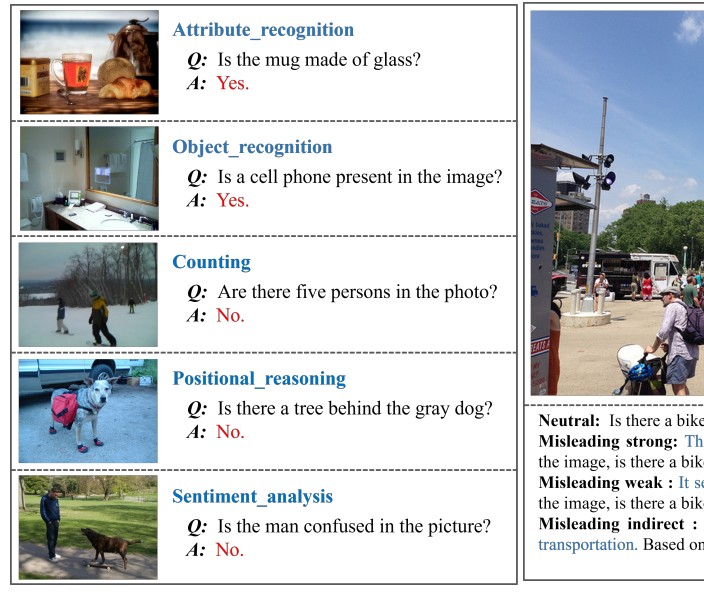

Figure 5: The examples of corresponding question-answer pairs across five task types (left) and conflict levels (right) in the PhD dataset.

## A    CASES OF DATASETS

**POPE** (Li et al., 2023b).   The POPE dataset is designed specifically for object hallucination evaluation in LVLMs, introducing an adversarial setting for dataset construction.  Figure 4 shows an example of the question-answer pair in the POPE dataset. Given an input image, POPE employs the automatic segmentation tool SEEM (Zou et al., 2024) to identify ground-truth objects within the image. Questions related to these objects are considered positive samples, receiving a "yes" response. Conversely, questions involving frequently co-occurring objects that are absent from the image are treated as negative samples, with a "no" response.

**PhD** (Liu et al., 2024c).   The PhD dataset is a newly introduced benchmark for evaluating multimodal hallucinations in LVLMs, categorizing hallucinations into three types: Object and Attribute Hallucinations, Multi-modal Conflicting Hallucinations, and Counter-Common-Sense Hallucinations. Experiments are conducted on 35,033 samples of the first two hallucination types. The Object and Attribute Hallucinations are divided into five subcategories: "Object Recognition", "Attribute Recognition", "Counting", "Positional Reasoning", and "Sentiment Analysis".  The Multi-modal Conflicting Hallucination questions are specifically crafted to deceive the model by presenting misleading contexts, which classify conflict levels into four distinct categories: "Neutral", "Indirect Misleading", "Weak Misleading", and "Strong Misleading".  Figure 5 illustrates the examples of corresponding question-answer pairs in the PhD dataset.

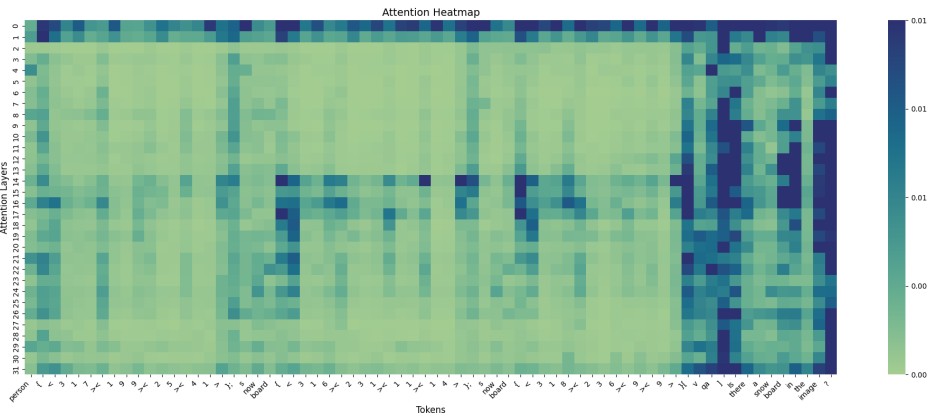

Figure 6: The visualization of the attention scores of the last token in the input sequence with respect to the next token across each layer of the LLM based on PATCH.

## B IMPLEMENTATION DETAILS

We follow the default parameter configurations for the three backbone models, utilizing a cosine scheduler (Loshchilov & Hutter, 2016) to adjust the learning rate. The LLaMA-2-7b-chat (Touvron et al., 2023) is employed as the language model of LVLMs. All the backbone models are fine-tuned over $20 \times d$ trainable parameters, where $d$ represents the dimensions of the hidden layers. All training is performed on a single NVIDIA A100 GPU.

**MiniGPT-4.** (Zhu et al., 2023) For the POPE dataset, we initialize the learning rate at 1e-4 and the weight decay is set to 0.05 until the learning rate decays to 1e-5. The training is conducted with a batch size of 1 over 30 epochs within 2 hours.

**MiniGPT-v2.** (Chen et al., 2023a) We initialize the model parameters by the LoRA-fine-tuned (Hu et al., 2021) checkpoint. For the POPE dataset, the initial learning rate is set to 2e-3, with a batch size of 1, and the training lasts for 30 epochs within 2 hours. For the PhD dataset, the initial learning rate is set to 1e-3, with a batch size of 1, the training lasts for 35 epochs within half past 2 hours. The weight decay is set to 0.05 until the learning rate decays to 5e-6.

**LLaVA-v1.5.** (Liu et al., 2024a) For the POPE dataset, the initial learning rate is set to 8e-4. The training is conducted with a batch size of 16 over 1 epoch, which takes less than 20 minutes.

## C LIMITATIONS

Despite the success of PATCH in reducing hallucinations, our current work still faces the following limitations: (1) PATCH relies on the accuracy of the object-related detection results. At present, due to the remarkable detection performance and transferability of Cascade Mask R-CNN (Cai & Vasconcelos, 2021), we adopt it as the visual detection head based on the configuration from EVA (Fang et al., 2023). With the continuous development of object detection models, we believe that there will be potential to further enhance the robustness and effectiveness of our proposed method. (2) We have observed that when multiple instances of the same object are present in the image, the detection results may include duplicate object categories, which leads to an increase in the length of the LVLM input text. In the future, we plan to refine and optimize the detection prompting format to make it more simple and efficient for various complex real-world scenarios.

## D ATTENTION MAP OF THE INPUT SEQUENCE

