# OpenReview forum: "From Pixels to Tokens: Revisiting Object Hallucinations in Large Vision-Language Models"
_ICLR.cc/2025/Conference — Submitted to ICLR 2025_

### Official Review · Reviewer_fcy9 · 2024-10-24

**Soundness:** 2
**Presentation:** 3
**Contribution:** 2
**Rating:** 5
**Confidence:** 4

**Summary:**

This paper revisits the reason for LVLM object hallucination by conducting a preliminary study and finding that visual feature decoupling instead of extraction is the main reason. Based on a potential solution, the authors propose PATCH, a novel plug-and-play prompt-tuning-based LVLM hallucination mitigation method. Experiments on the POPE and PhD datasets verify the effectiveness of the proposed method.

**Strengths:**

- Starting from a preliminary study and then introducing the proposed method, the writing is good.
- The proposed method is plug-and-play without requiring a large amount of computing.
- The authors demonstrate the effectiveness of the proposed methods on three baseline models.

**Weaknesses:**

- About the Cascade Mask-RCNN head:
  - Do you use a separate Cascade Mask R-CNN detector or just the detector head which is then connected with the vision encoder of the LVLM?
  - If the former, you cannot get the conclusion in Sec. 2 since the capabilities of the detector and the vision encoder of LVLM might differ from each other.
  - If the latter, how do you do that? Is the detection head exactly pre-trained with the same vision encoder of the LVLM, or do you need further training?

- About PATCH:
  - On the one hand, detection information might be redundant.
    - In lines 226-227, the authors suggest that,
      > In scenarios where no extra detection information is required, the LVLM can revert to processing the input using its standard capabilities without PATCH involvement.
    - I wonder how to do that, since the detection module of PATCH is query-irrelevant, suggesting that the detector will detect all objects in the image regardless of the question.
  - On the other hand, detection information is limited by the granularity of the detector.

    - PATCH aims at directly providing the object information detected by the detector to LLMs, which, however, cannot provide information that cannot be detected by detectors. For example, attributes (colors and shapes) cannot be detected, which are more common phenomenon of LVLM hallucinations nowadays.
    - Moreover, objects beyond the recognizable classes of the detector are also not recognizable.
  - Therefore, beyond experiment numbers, an analysis of why PATCH eases LVLM hallucination is also feasible. For example, will PATCH help if the queried object is *without* or *beyond* the class set of the pre-trained detector?

- About experiments:
  - Baselines: HA-DPO and HACL are both methods requiring training. Do you train them with the same training set with PATCH?
  - We care about hallucination, but we do not want to hurt the utility of LVLMs. Therefore, besides experiments on hallucination benchmarks like POPE and PhD, results on utility benchmarks (e.g., MME, MMBench, and SEED) are also important to demonstrate that hallucination mitigation is not at the cost of utility.

- Overall, I think this is an interesting paper, which, however, needs further work on 1) method clarification, 2) effectiveness analysis (i.e., what it can and cannot do), and 3) utility experiments.

**Questions:**

- About title:
  - The overall idea of this paper is to help LVLMs better utilize the object recognition information extracted by the vision encoder.
  - So what does the "From Pixels to Tokens" suggest in the title?
- About the potential solution:
  - How do you get the object-related information, by using the ground truths or prediction results of Cascade Mask R-CNN?
  - How do you deal with objects which are not existing in the image?

---

> ### Author Response · Authors · 2024-11-19
> **Response to Reviewer fcy9**
>
> Thank you for your valuable feedback and constructive comments. We would like to address your concerns in weaknesses and questions one by one as follows.
>
> **W1**: About the Cascade Mask-RCNN head.
>
> **Response**: We sincerely apologize if we did not explain this clearly enough. In our preliminary experiments, we combine the visual module of MiniGPT-v2 with a pre-trained Cascade Mask R-CNN **head** (i.e., the latter case you mentioned). In fact, [1] has provided the pre-trained parameters based on the joint training of the MiniGPT's visual encoder and the Cascade Mask R-CNN head, so we do not need to pre-train it by ourselves.
>
> In practical applications, users have the flexibility to leverage existing high-performance object detection models to detect object categories and their corresponding bounding boxes in images, thus mitigating the additional computational cost associated with training a dedicated object detector.
>
> **W2**: About PATCH.
>
> **Response**: Thank you for your valuable feedback.
>
> 1）Redundancy in detection information: The experimental results mentioned in the paper have demonstrated that PATCH can effectively integrate the embedding of objects into the prefill stage, improving errors related to attribute hallucination. However, when the query is unrelated to the objects in PATCH, redundancy may arise. We mitigate this redundancy through the following strategies: When the task is related to object attributes, we incorporate the PATCH module to enhance the model's focus on object-related details. Conversely, when the task is unrelated to object attributes, we exclude the PATCH module and utilize task-specific labels (i.g., [identify], [vqa], [refer]) mentioned by MiniGPT-V2 for model guiding.
>
> 2）We visualize the attention scores of the last token in the input sequence with respect to the next token across each layer of the LLM. The heatmap reveals that when querying the presence of objects in the detection information, specific embeddings of the objects exhibit higher scores, which indicates that our method effectively enhances semantic alignment through virtual tokens. The corresponding attention map is provided in Appendix D in the revised manuscript.
>
> 3）Objects beyond the recognizable classes and beyond experiment numbers: We have conducted ablation experiments by replacing the detected object categories with an generic placeholder "object{idx}". The experimental results have demonstrated a 5.36% accurancy improvement compared to the original baseline, which indicates that PATCH can also play a significant role in mitigating hallucinations, even in scenarios where the visual detection categories are uncertain. The detailed experimental results are below.
>
> **(1) origin:** zero-shot inference on MiniGPT-v2, prompt: \<Img>[Image]\</Img>[vqa][Question].
> **(2) PATCH:** fine-tuning on 20 virtual tokens on MiniGPT-v2, prompt: \<Img>[Image]\</Img>[Virtual_tokens]Objects:[Object][vqa][Question]. The object-related information is included at the [Object] position, formatted as "category\{<x1><x2><x3><x4>\}".
> **(3) PATCH + placeholder:** fine-tuning on 20 virtual tokens on MiniGPT-v2, prompt: \<Img>[Image]\</Img>[Virtual_tokens]Objects:[Object][vqa][Question]. The object-related information is included at the [Object] position, formatted as "category{idx}\{<x1><x2><x3><x4>\}".
>
> | Method | Accuracy | Precision | Recall | F1 |
> |:------|:----:|:------:|:------:|:------:|
> | （1）origin  | 83.37 | 88.00 | 77.27 | 82.29 |
> |（2） **PATCH** | **90.03** | **91.39** | **88.40** | **89.87** |
> | （3）PATCH + placeholder | 88.73 | 88.89 | 88.53 | 88.71 |
>
> **Reference**
> [1] Fang Y, Wang W, Xie B, et al. Eva: Exploring the limits of masked visual representation learning at scale[C]//Proceedings of the IEEE/CVF Conference on Computer Vision and Pattern Recognition. 2023: 19358-19369.

---

> ### Author Response · Authors · 2024-11-19
> **Response to Reviewer fcy9**
>
> **W3**: About experiments.
>   - Baselines: HA-DPO and HACL are both methods requiring training. Do you train them with the same training set with PATCH?
>   - We care about hallucination, but we do not want to hurt the utility of LVLMs. Therefore, besides experiments on hallucination benchmarks like POPE and PhD, results on utility benchmarks (e.g., MME, MMbench, and SEED) are also important to demonstrate that hallucination mitigation is not at the cost of utility.
> - Overall, I think this is an interesting paper, which, however, needs further work on 1) method clarification, 2) effectiveness analysis (i.e., what it can and cannot do), and 3) utility experiments.
>
> **Response**：
> 1）Baselines: Thank you for your detailed review. This is indeed an important issue that requires careful consideration, as differences in training set can impact the fairness of performance comparisons which is a common challenge in multi-modal hallucination tasks. In our experiments, we utilized the POPE (adversarial version) dataset constructed from the A-OKVQA dataset provided by [1] as our training set. In contrast, HA-DPO and HACL were trained on data generated by GPT-4, leading to differences in training datasets. To ensure fairness in comparisons, for hallucination mitigation methods requiring training, we directly reported the best results from their original papers. Furthermore, we plan to retrain HA-DPO and HACL on the same training set to provide a more direct comparison and will update the corresponding results in the revised version.
>
>
> 2）Influence on utility benchmarks: Thank you for raising this concern. In practice, for tasks where hallucination mitigation is not significantly impacted, users can opt to bypass PATCH and rely on the original model directly, thereby avoiding any potential negative effects introduced by the additional tokens. For tasks prone to hallucinations, such as VQA, PATCH can slightly enhance the performance, especially for tasks highly related to object attributes. We conducted relevant experiments on VQAv2, and the results are presented as follows. From the experimental results, we observe that our method improves performance on fundamental VLM tasks. Detailed experimental results will be further enhanced and provided in the revised version.
>
> **(1) origin:** zero-shot inference on MiniGPT-v2, prompt: \<Img>[Image]\</Img>[vqa][Question].
> **(2) PATCH:** fine-tuning on 20 virtual tokens on MiniGPT-v2, prompt: \<Img>[Image]\</Img>[Virtual_tokens]Objects:[Object][vqa][Question]. The object-related information is included at the [Object] position, formatted as "category\{<x1><x2><x3><x4>\}".
>
> | Method | Accuracy |
> |:------:|:----:|
> | （1）origin | 72.88 |
> | （2）**PATCH** | **73.16** |
>
>
> | Question Categories | origin Accuracy | PATCH Accuracy|
> |:------:|:----:|:------:|
> | what is this | 70.91 | **75.40** |
> | what are | 67.51 | **69.28** |
> | what sport is | 93.28 | **95.47** |
> | what animal is | 85.52 | **87.20** |
> | what is the name | 30.77 | **31.23** |
>
> **Reference**
> [1] Li Y, Du Y, Zhou K, et al. Evaluating object hallucination in large vision-language models[J]. arXiv preprint arXiv:2305.10355, 2023.

---

> ### Author Response · Authors · 2024-11-19
> **Response to Reviewer fcy9**
>
> **Q1**: About title.
>
> **Response**：Since this paper aims to explore the fundamental causes of multimodal hallucination, we investigate the visual encoder (feature extraction) and modality alignment module (feature disentanglement) from an architectural perspective, specifically the process from image pixels to LLM tokens. Therefore, we added "From Pixels to Tokens" to the title. Based on your suggestion, we will continue to consider whether a more suitable title might better reflect the scope of our work.
>
> **Q2**: About the potential solution.
>
> **Response**：
>
> 1）We obtain the object-related information by utilizing the prediction results of Cascade Mask R-CNN.
>
> 2）To address cases where the object names are uncertain, we conducted ablation experiments by replacing the detected object categories with an generic placeholder "object{idx}". The zero-shot experimental results have demonstrated a 2.66% accurancy drop compared to the original baseline, indicating that relying solely on generic placeholders is insufficient for optimal performance. While further combining it with PATCH has resulted in a 5.36% accuracy improvement, demonstrating the effectiveness of PATCH in enhancing the model's understanding ability, particularly in scenarios where the object names are uncertain in the image. For objects that are not present in the image, the visual detection head does not detect any corresponding objects, which makes it more likely for the LVLM to generate a "no" response.
>
> **(1) origin:** zero-shot inference on MiniGPT-v2, prompt: \<Img>[Image]\</Img>[vqa][Question].
> **(2) origin + placeholder:** zero-shot inference on MiniGPT-v2, promp)t: \<Img>[Image]\</Img>Objects:[Object][vqa][Question]. The object-related information is included at the [Object] position, formatted as "category{idx}\{<x1><x2><x3><x4>\}".
> **(3) PATCH:** fine-tuning on 20 virtual tokens on MiniGPT-v2, prompt: \<Img>[Image]\</Img>[Virtual_tokens]Objects:[Object][vqa][Question]. The object-related information is included at the [Object] position, formatted as "category\{<x1><x2><x3><x4>\}".
> **(4) PATCH + placeholder:** fine-tuning on 20 virtual tokens on MiniGPT-v2, prompt: \<Img>[Image]\</Img>[Virtual_tokens]Objects:[Object][vqa][Question]. The object-related information is included at the [Object] position, formatted as "category{idx}\{<x1><x2><x3><x4>\}".
>
> | Method | Accuracy | Precision | Recall | F1 |
> |:------|:----:|:------:|:------:|:------:|
> | (1) origin  | 83.37 | 88.00 | 77.27 | 82.29 |
> | (2) origin + placeholder | 80.70 | 86.40 | 72.87 | 79.06 |
> | (3) **PATCH** | **90.03** | **91.39** | **88.40** | **89.87** |
> | (4) PATCH + placeholder | 88.73 | 88.89 | 88.53 | 88.71 |

---

> > ### Comment · Reviewer_fcy9 · 2024-11-22
> > **Response to Author Reply**
> >
> > Thank you for your detailed responses. Here is my comment,
> >
> > - W2: about the usage of PATCH
> >   - According to the author's reply, whether or not the PATCH module is utilized during inference is dependent on the human prior on whether the object information will be helpful for the tasks or not, which limits the usage of PATCH, especially for zero-shot unseen tasks.
> >   - In other words, I can also interpret the experimental results as *selecting the benchmarks where PATCH is beneficial*.
> >   - Thus, it is important for the authors to specify,
> >     - How to define `In scenarios where no extra detection information is required` (line 226-227), and provide some example benchmarks,
> >     - Report the results of LVLMs with PATCH on these benchmarks to better observe the effect of PATCH.
> > - W3: about baselines
> >     - This is exactly why I ask about the training since it is unusual to train LVLMs on POPE and PhD, which are both commonly used evaluation datasets with specifically designed task formats (e.g., Yes/No for POPE).
> >     - Training on POPE is a huge advantage to the author's method, which is why only training with the same data can convince me of the effectiveness of the proposed method.
> > - W3: about utility
> >     - The problem remains. POPE comes from COCO, the same as VQAv2. Evaluating LVLMs with the same source of images is not as strong as evaluating commonly used benchmarks like MME, MMBench, and SEED.
> >
> > Overall, I do believe in the effectiveness of PATCH in carefully designed experimental settings, while a more throughout evaluation is necessary to convince me of the effectiveness for an open-world LVLM.

---

> > > ### Author Response · Authors · 2024-11-23
> > > **Response to Reviewer fcy9**
> > >
> > > Thanks for your further comments, which are very valuable for improving our work. We would like to try our best to address your concerns.
> > >
> > > **W2**: about the usage of PATCH
> > >
> > > **Response:** Thank you for questioning our approach, as it has prompted us to reflect further on the application scenarios of PATCH. Our study is specifically focused on **tackling the object hallucination problems in LVLMs**, which are most commonly observed in tasks like VQA and OCR. We need some prior knowledge about "whether the task contains object hallucinations" rather than "whether the object information will be helpful". If the task itself does not contain object hallucination problems or contains other problems, our method may not be suitable for application. Therefore, we think an ideal application case is that there is another object hallucination judgment model, which can provide delegate labels for the prior knowledge. While it is very interesting, we think it is beyond the scope of our method, and we would like to leave it as a future work.
> > >
> > > Additionally, POPE and PhD are two benchmarks highly relevant to object hallucination problems, which is why we specifically conducted experiments on these datasets and discussed them in detail in the manuscript.
> > >
> > >
> > > **W3**: about baselines
> > >
> > > **Response:** Thank you for your detailed review. This is indeed an important issue that requires careful consideration. To address this, we conducted an experiment where the virtual tokens fine-tuned on the POPE (AOKVQA-version) dataset were transferred to the PhD dataset for zero-shot inference. We have thoroughly verified that there is no distribution overlap between the POPE and PhD datasets to ensure the validity of our experimental setup. The results demonstrated a noticeable improvement, not only validating the effectiveness of our method in tackling object hallucination problems in LVLMs but also showcasing its plug-and-play capability. We hope the attached experimental results help address your concerns.
> > >
> > > **(1) origin:** zero-shot inference on MiniGPT-v2 on PhD dataset, prompt: \<Img>[Image]\</Img>[vqa][Question].
> > > **(2) origin + det:** zero-shot inference on MiniGPT-v2 on PhD dataset, prompt: \<Img>[Image]\</Img>Objects:[Object][vqa][Question]. The object-related information is included at the [Object] position, formatted as "category{idx}\{<x1><x2><x3><x4>\}".
> > > **(3) origin + det + PATCH(POPE(sft)):** zero-shot inference on MiniGPT-v2 on PhD dataset with PATCH, which has been fine-tuned on POPE(AOKVQA-version), prompt: \<Img>[Image]\</Img>[Virtual_tokens]Objects:[Object][vqa][Question]. The object-related information is included at the [Object] position, formatted as "category\{<x1><x2><x3><x4>\}".
> > >
> > > | Method | Accuracy | Precision | Recall | F1 |
> > > |:------|:----:|:------:|:------:|:------:|
> > > | (1) origin  | 47.92 | 52.46 | 55.48 | 53.93 |
> > > | (2) origin + det | 49.48 | 53.20 | 66.86 | 59.25 |
> > > | (3) **origin + det + PATCH(POPE(sft))** | **52.84** | **55.91** | **66.90** | **60.91** |
> > >
> > >
> > > **W3**: about utility
> > >
> > > **Response:** As with our response to W2, our study is specifically focused on **tackling the object hallucination problems in LVLMs** and identifying an effective solution to address this issue, rather than evaluating the effectiveness of our method in an open-world LVLM setting. In our previous response, we provided experimental results where the virtual tokens fine-tuned on the POPE (AOKVQA-version) dataset were transferred to the PhD dataset for MiniGPTv2's zero-shot inference. These results demonstrated the zero-shot transferability of our method and its effectiveness in mitigating object hallucination problems. We plan to explore a more general solution for hallucinations in open-world LVLMs in our future work.

---

> > > > ### Comment · Reviewer_fcy9 · 2024-11-27
> > > > **Response to Author Reply**
> > > >
> > > > Thank you for the detailed responses!
> > > >
> > > > I think we both understand that my main concern is about the effectiveness of PATCH for real-life LVLMs beyond academic benchmarks where,
> > > > 1. We know ahead of time about the query type (e.g., VQA or OCR).
> > > > 2. We know ahead of time that hallucinations will happen.
> > > > 3. We do not care how useful the response is as long as it does not hallucinate.
> > > >
> > > > Even if we do not expect a solution, I think it is essential to conclude first whether PATCH will harm the utility of LVLMs, which, however, seems contradicted in the authors' replies,
> > > > 1. In this [author reply](https://openreview.net/forum?id=ZPTHI3X9y8&noteId=Dxyaqz8ABD), the authors mention
> > > > > If the task itself does not contain object hallucination problems or contains other problems, our method may not be suitable for application.
> > > > 2.  In this [author reply](https://openreview.net/forum?id=ZPTHI3X9y8&noteId=9upNmSCFql), the authors mention PATCH will improve the performance on VQAv2.
> > > >
> > > > Before the aforementioned problems are solved, it is hard to convince me that the community will use PATCH in its own model, so I tend to retain my original score.

---

> > > > > ### Author Response · Authors · 2024-11-29
> > > > > **Response to Reviewer fcy9**
> > > > >
> > > > > Thank you for your patient response. We understand your concerns and will explain from the following aspects.
> > > > >
> > > > > We sincerely apologize for any confusion caused by our previous response. Our previous response was intended to convey that, our method is particularly effective in addressing tasks prone to object hallucination. However, it may not be as suitable for tasks that do not inherently involve object hallucinations. To enhance its applicability, we believe incorporating an object hallucination detection model would be beneficial. Such a model could help identify whether a user's query is likely to involve object hallucinations, thereby allowing PATCH to achieve even greater effectiveness. In the following response, we present the experimental results of our method on the MME Benchmark which demonstrates that PATCH not only does not impaire the LVLM's overall capability, but also contributes significantly to perception tasks.
> > > > >
> > > > > **Performance of PATCH on General Benchmark：**
> > > > >
> > > > > Thanks to ICLR for giving us 6 extra days for discussion, which provided us with the opportunity to conduct further experiments on the General Benchmark. We carried out experiments on MME, and the specific experimental results are shown in the tables below：
> > > > >
> > > > > **(1) origin:** zero-shot inference on MiniGPT-v2 on MME, prompt: \<Img>[Image]\</Img>[vqa][Question].
> > > > > **(2) origin + det:** zero-shot inference on MiniGPT-v2 on MME, prompt: \<Img>[Image]\</Img>Objects:[Object][vqa][Question]. The object-related information is included at the [Object] position, formatted as "category{idx}\{<x1><x2><x3><x4>\}".
> > > > > **(3) origin + det + PATCH(POPE(sft)):** zero-shot inference on MiniGPT-v2 on MME with PATCH, which has been fine-tuned on POPE(AOKVQA-version), prompt: \<Img>[Image]\</Img>[Virtual_tokens]Objects:[Object][vqa][Question]. The object-related information is included at the [Object] position, formatted as "category\{<x1><x2><x3><x4>\}".
> > > > >
> > > > > |Methods| Perception | Existence | Count | Positon | Color | Poster | Celebrity | Scence | Landmark | Artwork | OCR |
> > > > > |:------|:----:|:------:|:------:|:------:|:------:|:------:|:------:|:------:|:------:|:------:|:------:|
> > > > > | (1) origin  | 1317.45 | **195** | **131.67** | 81.67 | 130 | 149.66 | **144.71** | 152.25 | 113.5 | **134** | 85 |
> > > > > | (2) origin + det | 1290.96 | **195** | 130 | 73 | 125 | 147.28 | 136.18 | 153.75 | 126.6 | 131.75 | 72.5 |
> > > > > | (3) **origin + det + PATCH(POPE(sft))** | **1352.2** | **195** | 130 | **83.33** | **135** | **153.4** | 136.47 | **153.75** | **133.25** | 132 | **100** |
> > > > >
> > > > > |Methods| Cognition | Commonsense Reasoning | Numerical Calculation | Text Translation | Code Reasoning |
> > > > > |:------|:----:|:------:|:------:|:------:|:------:|
> > > > > | (1) origin  | **269.64** | **107.14** | **50** | **50** | 62.5 |
> > > > > | (2) origin + det | 260.71 |  90.71 | **50** | **50** | **70** |
> > > > > | (3) **origin + det + PATCH(POPE(sft))** | 261.43 | 91.43 | **50** | **50** | **70** |
> > > > >
> > > > > The evaluation tasks in MME can be categorized into Perception and Cognition tasks. The experimental results demonstrated that our method does not negatively impact the general performance of the LVLM. In fact, PATCH achieved a 34.55-point improvement in Perception task (i.e., Position, Color, Poster, Scene, Landmark, OCR) compared to Minigpt-v2. Although, there was an 8.21-point decrease in Cognition tasks, since PATCH was only fine-tuned on the POPE dataset, we believe our method could achieve better generalization through further training on other large-scale datasets.

---

> ### Author Response · Authors · 2024-11-22
> **To confirm whether our responses has addressed the concerns and thank you for your efforts**
>
> Dear Reviewer,
>
> We sincerely appreciate your time and effort in reviewing our paper and offering valuable suggestions. Since there will be no second stage of author-reviewer discussions, and the discussion phase is drawing to a close, we would to confirm whether our responses have effectively addressed your concerns a few days ago. And we hope they have adequately addressed your issues. If you require further clarification or have additional concerns, please don’t hesitate to contact us. We are more than willing to continue our communication with you.
>
> Best regards.

---

### Official Review · Reviewer_84zh · 2024-10-24

**Soundness:** 3
**Presentation:** 3
**Contribution:** 3
**Rating:** 8
**Confidence:** 5

**Summary:**

The paper investigates the root causes of hallucinations in LVMs, identifying the main issue as the insufficient separation of textual and visual features during multi-modal integration. To address this, the authors introduce a tuning-based method, designed to help LVMs leverage object detection to reduce hallucinations. The effectiveness of this method is demonstrated through validation on two multi-modal hallucination evaluation datasets across three different LVMs.

**Strengths:**

1. The paper is very clear and easy to follow.

2. The experiments support the main claims in the text, and the presented approach is simple and useful.

3. Most of the design choices are ablated to verify their effectiveness.

**Weaknesses:**

1. The paper claims that the reason for hallucinations is the feature decoupling. Nevertheless, many of the hallucinations in Table 1 are also not detected by the detection model (308 vs. 191). This suggests that the encoder is an additional significant source of hallucinations. I would like to see the same table, after the replacement of $prompt_1$ with $prompt_2$.

2. The paper claims that the additional computational cost of training the extra tokens is negligible. Nevertheless, the paper omits the fact that the model relies on an additional object detector that should also be trained. It is not clear to me if the additional computational cost and model parameters that were spent on the object detector, can be spent instead on a larger encoder/adapter/llm, and result in a much cleaner solution for hallucination reduction.

3. Overall, the paper presents a simple soft-prompting approach for targeting hallucination reduction. Are the detections necessary, or can they be achieved solely from additional soft-prompting with more learned tokens?

**Questions:**

1. For $prompt_2$ and along the rest of the paper, the additional information that is extracted from the object detector and given to the language model includes not only the existence of objects but also their location (x1,x2,y1,y2). Although according to Table 4, it is necessary, it is not clear why (the VQA is about object existence and not localization). Do you have any intuition for this property? Can you compare the accuracy of the bboxes to their hallucination removal rate?

2. Is the VLM even needed, given the soft prompt and the detection results? What is the baseline of soft-prompting a text-only LLM on the object detection results, without the use of images?

---

> ### Author Response · Authors · 2024-11-19
> **Response to Reviewer 84zh**
>
> Thank you for your valuable feedback and constructive comments. We would like to address your concerns in weaknesses and respond to your questions one by one as below.
>
> **W1**: The encoder is an additional significant source of hallucinations. Need results after the replacement of prompt$_1$ with prompt$_2$.
>
> **Response**: Thank you for your valuable feedback. The results in Table 1 suggest that 74.58% hallucinations occur when object detection is accurate while the model’s inference is incorrect, indicating that in the current experiment, the modal alignment module is the **primary** cause of hallucinations, with a frequency ratio of 3:1 compared to the visual encoder (we believe this ratio would be even higher if a more advanced detector head is used).
>
> |  Results  | Correct Inf. | Wrong Inf. |
> |:------|:----:|:------:|
> | Correct Detection | 2670 | 97 |
> | Wrong Detection | 56 | 240 |
>
> Additionally, we conducted experiments with the replacement of prompt$_1$ with prompt$_2$. A comparison of Prompt2's inference results on MiniGPT-v2 with the detection results revealed that incorporating object information into the LLM effectively mitigates multi-modal hallucinations. Furthermore, the tabular results demonstrate consistent behavior between inference errors and detection errors after adding object-related information. We attribute these errors to the limitations of the detector's performance, suggesting that using a more advanced detector could further enhance the performance of our method.
>
> **W2**: The model relies on an additional object detector that should also be trained.
>
> **Response**: Thank you for raising this concern. In our experiments, we employed a pre-trained object detection head provided by [1] for object recognition. Consequently, the training cost of this component was not included in the computational cost reported for our method. Notably, MiniGPT-v2 has a total of 7,748,467,072 parameters, while the detection head contains only 48,527,951 trainable parameters. Given the significantly smaller parameter count of the detection head compared to the overall model, training it would not impose a substantial computational burden. Moreover, in practical applications, users have the flexibility to leverage existing high-performance object detection models to detect object categories and their corresponding bounding boxes in images, thus mitigating the additional computational cost associated with training a dedicated object detector.
>
> We appreciate your insightful suggestion about reallocating the computational cost of training an object detector towards enlarging the encoder, adapter, or LLM to potentially achieve a more streamlined solution for reducing hallucinations. This is indeed an intriguing direction for future research, and we plan to explore this aspect further in our future work.
>
> **W3**: Are the detections necessary, or can they be achieved solely from additional soft-prompting with more learned tokens?
>
> **Response**：Thanks for posting this valuable question. We conducted a token number ablation experiment where only virtual tokens were added **without** incorporating the detection results. The experimental results are shown below.
>
> **(1) origin:** zero-shot inference on MiniGPT-v2, prompt: \<Img>[Image]\</Img>[vqa][Question].
> **(2) + k token:** fine-tuning k virtual tokens on MiniGPT-v2, prompt: \<Img>[Image]\</Img>[Virtual_tokens][vqa][Question].
>
> |  Method  | Accuracy | Precision | Recall | F1 |
> |:------|:----:|:------:|:------:|:------:|
> | **origin**  | **83.37** | **88.00** | **77.27** | **82.29** |
> | + 20 token | 82.60 | 84.93 | 79.27 | 82.00 |
> | + 30 token | 82.27 | 82.97 | 81.20 | 82.08 |
> | + 50 token | 81.40 | 78.72 | 86.07 | 82.23 |
> | + 100 token | 77.67 | 73.29 | 87.07 | 79.59 |
>
> The results clearly demonstrate that the inclusion of detection results is crucial, which indicates that merely increasing the number of trainable prompting tokens is insufficient for mitigating hallucinations in LVLMs. Additionally, increasing the number of virtual tokens raises the fine-tuning costs for the model, which in turn results in a gradual decline in performance when the learning rate and number of training epochs remain fixed.
>
> **Reference**
> [1] Fang Y, Wang W, Xie B, et al. Eva: Exploring the limits of masked visual representation learning at scale[C]//Proceedings of the IEEE/CVF Conference on Computer Vision and Pattern Recognition. 2023: 19358-19369.

---

> ### Author Response · Authors · 2024-11-19
> **Response to Reviewer 84zh**
>
> **Q1** Why localization information is important for VQA? Can you compare the accuracy of the bboxes to their hallucination removal rate?
>
> **Response**：We conducted ablation experiments by replacing the detected object categories with an generic placeholder "object{idx}". The zero-shot experimental results have demonstrated a 2.66% accurancy drop compared to the original MiniGPT-v2, indicating that relying solely on generic placeholders is insufficient for optimal performance. By observing the third and fourth rows, we can find that adding either only object category information or only object location information allows PATCH to partially mitigate hallucinations in LVLMs. Notably, providing object location information proves to be more effective. This may be attributed to the fact that detecting box information can help LVLMs achieve precise visual localization, thereby ensuring the faithful responses to the input image.
>
> **(1) origin:** zero-shot inference on MiniGPT-v2, prompt: \<Img>[Image]\</Img>[vqa][Question].
> **(2) origin + placeholder:** zero-shot inference on MiniGPT-v2, prompt: \<Img>[Image]\</Img>Objects:[Object][vqa][Question]. The object-related information is included at the [Object] position, formatted as "category{idx}\{<x1><x2><x3><x4>\}".
> **(3) PATCH:** fine-tuning on 20 virtual tokens on MiniGPT-v2, prompt: \<Img>[Image]\</Img>[Virtual_tokens]Objects:[Object][vqa][Question]. The object-related information is included at the [Object] position, formatted as "category\{<x1><x2><x3><x4>\}".
> **(4) PATCH + placeholder:** fine-tuning on 20 virtual tokens on MiniGPT-v2, prompt: \<Img>[Image]\</Img>[Virtual_tokens]Objects:[Object][vqa][Question]. The object-related information is included at the [Object] position, formatted as "category{idx}\{<x1><x2><x3><x4>\}".
> **(5) PATCH - bboxes:** fine-tuning on 20 virtual tokens on MiniGPT-v2, prompt: \<Img>[Image]\</Img>[Virtual_tokens]Objects:[Object][vqa][Question]. The object-related information is included at the [Object] position, formatted as "category".
>
>
> | Method | Accuracy | Precision | Recall | F1 |
> |:------:|:----:|:------:|:------:|:------:|
> | （1）origin  | 83.37 | 88.00 | 77.27 | 82.29 |
> | （2）origin + placeholder | 80.70 | 86.40 | 72.87 | 79.06 |
> | （3）**PATCH** | **90.03** | **91.39** | **88.40** | **89.87** |
> | （4）PATCH + placeholder | 88.73 | 88.89 | 88.53 | 88.71 |
> | （5）PATCH - bboxes | 88.27 | 87.37 | 89.47 | 88.41 |
>
> **Q2**: Is the VLM even needed, given the soft prompt and the detection results? What is the baseline of soft-prompting a text-only LLM on the object detection results, without the use of images?
>
> **Response**：We conducted an ablation experiments on MiniGPT-v2's language model (LLaMA2) to evaluate the baseline performance of prompting a text-only LLM solely based on object detection results without incorporating images. The experimental results indicate that relying solely on object detection prompts makes it challenging for the text-only model to infer the existence of objects in the image. Furthermore, as the amount of introduced information increases, the model's ability to effectively utilize relevant information diminishes. This finding further validates that our proposed PATCH method effectively enhances the LLM in LVLMs, enabling it to better leverage the valuable information contained in the input detection results.
>
> **(1) prompt1:** zero-shot inference on Llama2, prompt: Based on the detection results: [Object], [Question]. The object-related information is included at the [Object] position, formatted as "category".
> **(2) prompt2:** zero-shot inference on Llama2, prompt: Based on the detection results: [Object], [Question]. The object-related information is included at the [Object] position, formatted as "category\{<x1><x2><x3><x4>\}".
>
>
> | Prompts | Accuracy | Precision | Recall | F1 |
> |:------:|:----:|:------:|:------:|:------:|
> | （1）prompt1 | 75.77 | 72.28 | 83.60 | 77.53 |
> | （2）prompt2 | 68.70 | 63.81 | 86.40 | 73.41 | 67.70

---

> ### Author Response · Authors · 2024-11-22
> **To confirm whether our responses has addressed the concerns and thank you for your efforts**
>
> Dear Reviewer,
>
> We sincerely appreciate your time and effort in reviewing our paper and offering valuable suggestions. Since there will be no second stage of author-reviewer discussions, and the discussion phase is drawing to a close, we would to confirm whether our responses have effectively addressed your concerns a few days ago. And we hope they have adequately addressed your issues. If you require further clarification or have additional concerns, please don’t hesitate to contact us. We are more than willing to continue our communication with you.
>
> Best regards.

---

> > ### Comment · Reviewer_84zh · 2024-11-22
> >
> > I thank the authors for the rebuttal. The rebuttal addressed most of my concerns. I raised my score.

---

> > > ### Author Response · Authors · 2024-11-23
> > > **Thanks for the reply**
> > >
> > > We sincerely thank you for your recognition of this paper and your valuable suggestions. We’re glad we could address your concerns.

---

### Official Review · Reviewer_TZeq · 2024-10-28

**Soundness:** 2
**Presentation:** 2
**Contribution:** 2
**Rating:** 5
**Confidence:** 4

**Summary:**

The paper presents a parameter-efficient tuning technique that introduces detection for hallucination mitigation in multimodal large language models. The method utilizes adaptive virtual tokens to extract object features from bounding boxes, thereby addressing hallucinations caused by insufficient decoupling of visual features.

**Strengths:**

Computational Efficiency: The proposed PATCH strategy achieves parameter-efficient tuning by freezing LVLM parameters and optimizing a small number of virtual tokens, making it lightweight and practical for deployment scenarios.

**Weaknesses:**

1.	Lack of Clarity in Preliminary Experiments: Section 2 is confusing, particularly regarding Table 1. Definitions of “correct detection” and “wrong detection,” along with their calculation methods, are not adequately explained.
2.	Insufficient Support in Analysis: The performance gap between detection and inference is not convincingly attributed to the struggles of the visual projection module. The reasoning lacks robust experimental evidence.
3.	Contradiction in Motivation: While the motivation suggests that incorporating detection information may introduce redundancy, the method still relies on detection information plus soft visual tokens, creating a contradiction.
4.	Limited Novelty: The method primarily involves parameter-efficient tuning (soft prompt tuning), which does not introduce significant novelty beyond existing tuning techniques.
5.	Unfair Experimental Setup: The use of a portion of the test data for training gives an in-domain testing advantage, compromising the fairness of comparisons with baselines.
6.	Incomplete Baseline Comparisons: Not all baselines are consistently compared across different LVLMs, particularly in the main results table, diminishing the thoroughness of the evaluation.
7.	Omission of Experimental Details: Important experimental details, such as decoding strategies, are missing, limiting the reproducibility of the experiments.
8.	Absence of Ablation Studies: An ablation study isolating the added visual tokens is needed to properly assess their contribution to the model's performance. (e.g., tuning only the projector on the same data)
9.	Overclaimed Results: Some claims, such as those in lines 316–317 about enhanced semantic alignment via visual tokens, are overstated without sufficient empirical support.
10.	Incomplete Related Work: The related works section neglects recent advances in LVLMs, such as Qwen2VL and LLaVA-OneVision, as well as relevant hallucination mitigation methods like GAVIE, limiting the contextual relevance of the paper.

**Questions:**

Please refer to the weakness section.

---

> ### Author Response · Authors · 2024-11-19
> **Response to Reviewer TZeq**
>
> Thank you for your valuable feedback and thoughtful comments. We would like to address your concerns in weaknesses one by one as follows.
>
> **W1**: Lack of Clarity in Preliminary Experiments.
>
> **Response**: We are sorry for the unclear description. In our preliminary experiments, we combine the visual module of MiniGPT-v2 with a pre-trained Cascade Mask R-CNN head, which serves as the visual detection model for object recognition. When the detection model recognizes the object present the positive sample (i.e., the answer of the sample is "yes") or does not detect the object in the negative examples (i.e., the answer of the sample is "no"), we define the result as a "correct detection". In other cases, we define the result as a "wrong detection". We will add more explanation on this in our revised paper.
>
>
> **W2**: Insufficient Support in Analysis.
>
> **Response**: Thank you for your thoughtful feeaback. As mentioned in the paper, we hypothesize that multi-modal object hallucinations in LVLMs arise from two main sources:
> 1) Insufficient extraction of visual features: The visual encoder may fail to capture critical details or misinterpret objects in the image, resulting in inaccurate or incomplete visual representations.
> 2) Inadequate decoupling of visual features: Even when the visual encoder generates accurate features, the visual projection layer may struggle to align these features correctly with the corresponding textual embeddings in the LLM's semantic space.
>
> By observing the first row and first column in Table 1 mentioned in our article, we can see that the number of samples with correct detection results exceeds the number of samples with correct inference results. This suggests that the potential of the visual encoder for image understanding has not been fully leveraged by the language model of LVLMs. Consequently, we attribute the primary cause of hallucination issues in LVLMs to the inadequate decoupling of visual features.
>
> If you have any further suggestions, please feel free to share them with us. We would be glad to address them by conducting additional experiments to provide a thorough explanation.
>
> **W3**: Contradiction in Motivation.
>
> **Response**: Thank you for raising this concern. In our article, redundancy refers to instances where the detected object has weak relevance to the given question, rendering the object embeddings redundant. In such cases, PATCH assists the language model of the LVLM in filtering and optimizing the use of detection information. By employing trainable and pluggable virtual tokens as a bridge, PATCH enables LVLMs to focus on task-relevant image features, thereby improving the alignment between visual and textual representations within the semantic space. We will provide further clarification on this in the camera-ready version to ensure better understanding for the readers.
>
> **W4**: Limited Novelty.
>
> **Response**: Thanks for your feedback. The primary contribution of our research lies in identifying the specific module responsible for hallucination generation in LVLMs, rather than proposing a new lightweight fine-tuning method for enhancing the LVLM performance.
>
> Our preliminary experiments reveal that hallucinations primarily arise in the modal alignment module. To address this issue, we propose the PATCH strategy to enhance the alignment between visual and textual features. PATCH is a plug-and-play method that can be dynamically adjusted based on the requirements of specific applications. Furthermore, we have compared our method with several influential prior approaches, and our experimental results consistently demonstrate superior performance.
>
> **W5**: Unfair Experimental Setup.
>
> **Response**: We are sorry for the unclear description. We use the POPE (adversarial version) dataset constructed from the A-OKVQA dataset provided by [1] as our training set. This dataset is identical to the POPE test set only in terms of **data format**. Since the training set is separate from the test set, the fairness of the baseline comparison can be ensured.
>
> **W6**: Incomplete Baseline Comparisons.
>
> **Response**: We understand your concern. We use the POPE (adversarial version) dataset constructed from the MSCOCO dataset provided by [1] as our test set, which is also the test set for all the mentioned baseline methods. To ensure fairness in the comparison, for hallucination mitigation methods that require training, we directly used the experimental results from the original papers.
>
> **Reference**
> [1] Li Y, Du Y, Zhou K, et al. Evaluating object hallucination in large vision-language models[J]. arXiv preprint arXiv:2305.10355, 2023.

---

> ### Author Response · Authors · 2024-11-19
> **Response to Reviewer TZeq**
>
> **W7**: Omission of Experimental Details.
>
> **Response**: We apologize if our explanation was not very clear. In our experiments, we did not modify the decoding strategy, and all methods consistently use the **greedy decoding** strategy. These details will be provided in the revised version of our paper.
>
> **W8**: Absence of Ablation Studies.
>
> **Response**: We sincerely apologize for the unclear description. In our experiments, we **did not** fine-tune the language modeling head or the visual projection layer. Instead, only the virtual tokens on the encoding side were trained during the training phase. We conducted a thorough ablation study in the paper, and for improved clarity, we provide the results of all ablation experiments below. All ablation experiments were conducted based on MiniGPT-v2.
>
> **(1) origin:** zero-shot inference on MiniGPT-v2, prompt: \<Img>[Image]\</Img>[vqa][Question].
> **(2) origin + objs:** zero-shot inference on MiniGPT-v2, prompt: \<Img>[Image]\</Img>Objects:[Object][vqa][Question]. The object-related information is included at the [Object] position, formatted as "category".
> **(3) origin + objs & bboxes:** zero-shot inference on MiniGPT-v2, prompt: \<Img>[Image]\</Img>Objects:[Object][vqa][Question]. The object-related information is included at the [Object] position, formatted as "category\{<x1><x2><x3><x4>\}".
> **(4) origin(sft) + objs & bboxes:** LoRA fine-tuning on MiniGPT-v2, prompt: \<Img>[Image]\</Img>Objects:[Object][vqa][Question]. The object-related information is included at the [Object] position, formatted as "category\{<x1><x2><x3><x4>\}".
> **(5) PATCH:** fine-tuning on 20 virtual tokens on MiniGPT-v2, prompt: \<Img>[Image]\</Img>[Virtual_tokens]Objects:[Object][vqa][Question]. The object-related information is included at the [Object] position, formatted as "category\{<x1><x2><x3><x4>\}".
> **(6) PATCH - bboxes:** fine-tuning on 20 virtual tokens on MiniGPT-v2, prompt: \<Img>[Image]\</Img>[Virtual_tokens]Objects:[Object][vqa][Question]. The object-related information is included at the [Object] position, formatted as "category".
> **(7) PATCH - bboxes & objs:** fine-tuning on 20 virtual tokens on MiniGPT-v2, prompt: \<Img>[Image]\</Img>[Virtual_tokens][vqa][Question].
>
> Among these expeirments, (1)-(3) are proposed to evaluate the impact of different formats of detection results on MiniGPT-v2's inference. (4) is proposed to compare the effects between two different fine-tuning methods: virtual tokens and LoRA. (5)-(7) are proposed to evaluate the impact of different formats of detection results on PATCH.
>
> |  Method  | Accuracy | Precision | Recall | F1 |
> |:------|:----:|:------:|:------:|:------:|
> | (1) origin  | 83.37 | 88.00 | 77.27 | 82.29 |
> | (2) origin + objs | 88.80 | 88.19 | 89.60 | 88.89 |
> | (3) origin + objs \& bboxes | 89.33 | 89.81 | 88.73 | 89.27 |
> | (4) origin(sft) + objs \& bboxes | 87.97 | 88.45 | 87.33 | 87.89 |
> | (5) **PATCH** | **90.03** | **91.39** | **88.40** | **89.87** |
> | (6) PATCH - bboxes  | 88.27 | 87.37 | 89.47 | 88.41 |
> | (7) PATCH - bboxes \& objs | 82.60 | 84.93 | 79.27 | 82.00 |
>
> **W9**: Overclaimed Results Some claims.
>
> **Response**: Thanks for raising this concern. First, the results of Table 3 mentioned in our article demonstrate that our method outperforms other approaches on the POPE dataset, achieving superior performance. Second, we visualize the attention scores of the last token in the input sequence with respect to the next token across each layer of the LLM. The heatmap reveals that when querying the presence of objects in the detection information, specific embeddings of the objects exhibit higher scores, which indicates that our method effectively enhances semantic alignment through virtual tokens. The corresponding attention map is provided in Appendix D in the revised manuscript.

---

> ### Author Response · Authors · 2024-11-19
> **Response to Reviewer TZeq**
>
> **W10**: Incomplete Related Work.
>
> **Response**：Thanks for your advice. In the revised version, we will update the related work section to include the latest developments in LVLMs. Additionally, we will add more comparisons with related work in the experimental section (as shown below).
>
> |  Method  | Accuracy | Precision | Recall | F1 |
> |:------|:----:|:------:|:------:|:------:|
> | SoK[1] | 75.13 | 90.19 | 56.40 | 69.40 |
> | Groundhog[2] | 86.33 | 85.93 | 86.63 | 86.28 |
> | Qwen2-VL-7B[3] | 86.83 | 88.02 | 85.27 | 86.62 |
> | LLaVA OneVision-7B[4] | 87.13 | 90.25 | 83.27 | 86.62 |
> | GAVIE[5] | 62.30 | 57.46 | 94.66 | 71.51 |
> | LLaVA1.5+VCD[6] | 80.88 | 79.45 | 83.29 | 81.33 |
> | Chat-UniVi-7B[7] | 64.97 | - | - | 71.54|
> | Chat-UniVi-7B-v1.5[7] | 83.23 | - | - | 82.31 |
> | GoundingGPT[8] | 86.17 | - | - | 85.50 |
> | Osprey[9] | 85.33 | 85.43 | 85.20 | 85.31 |
> | GLaMM[10] | 87.30 | 85.03 | 90.53 | 87.70 |
> | **PATCH** | **90.03** | **91.39** | **88.40** | **89.87** |
>
> **Reference**
> [1] Yang, Jianwei, et al. "Set-of-mark prompting unleashes extraordinary visual grounding in gpt-4v." arXiv preprint arXiv:2310.11441 (2023).
> [2] Zhang Y, Ma Z, Gao X, et al. Groundhog: Grounding large language models to holistic segmentation[C]//Proceedings of the IEEE/CVF conference on computer vision and pattern recognition. 2024: 14227-14238.
> [3] Wang, Peng, et al. "Qwen2-vl: Enhancing vision-language model's perception of the world at any resolution." arXiv preprint arXiv:2409.12191 (2024).
> [4] Li, Bo, et al. "Llava-onevision: Easy visual task transfer." arXiv preprint arXiv:2408.03326 (2024).
> [5] Liu, Fuxiao, et al. "Mitigating hallucination in large multi-modal models via robust instruction tuning." The Twelfth International Conference on Learning Representations. 2023.
> [6] Leng, Sicong, et al. "Mitigating object hallucinations in large vision-language models through visual contrastive decoding." Proceedings of the IEEE/CVF Conference on Computer Vision and Pattern Recognition. 2024.
> [7] Jin, Peng, et al. "Chat-univi: Unified visual representation empowers large language models with image and video understanding." Proceedings of the IEEE/CVF Conference on Computer Vision and Pattern Recognition. 2024.
> [8] Li, Zhaowei, et al. "Groundinggpt: Language enhanced multi-modal grounding model." Proceedings of the 62nd Annual Meeting of the Association for Computational Linguistics (Volume 1: Long Papers). 2024.
> [9] Yuan, Yuqian, et al. "Osprey: Pixel understanding with visual instruction tuning." Proceedings of the IEEE/CVF Conference on Computer Vision and Pattern Recognition. 2024.
> [10] Rasheed, Hanoona, et al. "Glamm: Pixel grounding large multimodal model." Proceedings of the IEEE/CVF Conference on Computer Vision and Pattern Recognition. 2024.

---

> ### Author Response · Authors · 2024-11-22
> **To confirm whether our responses has addressed the concerns and thank you for your efforts**
>
> Dear Reviewer,
>
> We sincerely appreciate your time and effort in reviewing our paper and offering valuable suggestions. Since there will be no second stage of author-reviewer discussions, and the discussion phase is drawing to a close, we would to confirm whether our responses have effectively addressed your concerns a few days ago. And we hope they have adequately addressed your issues. If you require further clarification or have additional concerns, please don’t hesitate to contact us. We are more than willing to continue our communication with you.
>
> Best regards.

---

> > ### Comment · Reviewer_TZeq · 2024-11-23
> >
> > I have carefully read your responses and I appreciate your explanation and experiments. Your rebuttal helps clarify a few points that confused me, but I'm more concerned about the novelty of the proposed method, the validity of the experimental designs, as well as the used baselines and datasets in experiments.

---

> > > ### Author Response · Authors · 2024-11-23
> > > **Response to Reviewer TZeq**
> > >
> > > Thank you for your feedback. We are committed to addressing your concerns to the best of our ability. If you have any specific questions or issues, please feel free to let us know.

---

> > > ### Author Response · Authors · 2024-11-29
> > > **Response to Reviewer TZeq**
> > >
> > > Thanks to ICLR for giving us 6 extra days for discussion, which provided us more time to carefully reconsider and address your concerns.
> > >
> > > **The novelty of our method:** Firstly, we explore the intrinsic sources of hallucinations in LVLMs, revealing that the inadequate decoupling of textual and visual features during multi-modal integration is the primary cause of hallucinations
> > > in LVLMs. Then, we propose a plug-and-play method, called PATCH, to address the object hallucination issue in LVLMs, which leverages the virtual tokens and detection information to assist in the alignment of visual and textual features. Compared to other hallucination mitigation approaches, our method only requires a few computational costs for fine-tuning and is flexible to be  dynamically applied based on different application needs.
> > >
> > > **The effectiveness of our method:** We first conducted experiments on two publicly available multi-modal hallucination evaluation datasets: POPE and PhD. By comparing our method with others, the experimental results have shown the effectiveness of PATCH in mitigating object hallucinations in LVLMs. Then, we conducted a series of ablation experiments, as detailed in the response to Weakness 8, to validate the soundness of PATCH's design. Finally, we performed further experiments on a comprehensive MLLM evaluation benchmark (MME). The experimental results are shown in the tables below:
> > >
> > > **(1) origin:** zero-shot inference on MiniGPT-v2 on MME, prompt: \<Img>[Image]\</Img>[vqa][Question].
> > > **(2) origin + det:** zero-shot inference on MiniGPT-v2 on MME, prompt: \<Img>[Image]\</Img>Objects:[Object][vqa][Question]. The object-related information is included at the [Object] position, formatted as "category{idx}\{<x1><x2><x3><x4>\}".
> > > **(3) origin + det + PATCH(POPE(sft)):** zero-shot inference on MiniGPT-v2 on MME with PATCH, which has been fine-tuned on POPE(AOKVQA-version), prompt: \<Img>[Image]\</Img>[Virtual_tokens]Objects:[Object][vqa][Question]. The object-related information is included at the [Object] position, formatted as "category\{<x1><x2><x3><x4>\}".
> > >
> > > |Methods| Perception | Existence | Count | Positon | Color | Poster | Celebrity | Scence | Landmark | Artwork | OCR |
> > > |:------|:----:|:------:|:------:|:------:|:------:|:------:|:------:|:------:|:------:|:------:|:------:|
> > > | (1) origin  | 1317.45 | **195** | **131.67** | 81.67 | 130 | 149.66 | **144.71** | 152.25 | 113.5 | **134** | 85 |
> > > | (2) origin + det | 1290.96 | **195** | 130 | 73 | 125 | 147.28 | 136.18 | 153.75 | 126.6 | 131.75 | 72.5 |
> > > | (3) **origin + det + PATCH(POPE(sft))** | **1352.2** | **195** | 130 | **83.33** | **135** | **153.4** | 136.47 | **153.75** | **133.25** | 132 | **100** |
> > >
> > >
> > > |Methods| Cognition | Commonsense Reasoning | Numerical Calculation | Text Translation | Code Reasoning |
> > > |:------|:----:|:------:|:------:|:------:|:------:|
> > > | (1) origin  | **269.64** | **107.14** | **50** | **50** | 62.5 |
> > > | (2) origin + det | 260.71 |  90.71 | **50** | **50** | **70** |
> > > | (3) **origin + det + PATCH(POPE(sft))** | 261.43 | 91.43 | **50** | **50** | **70** |
> > >
> > > The evaluation tasks in MME can be categorized into Perception and Cognition tasks. The experimental results demonstrated that our method does not negatively impact the general performance of the LVLM. In fact, PATCH achieved a 34.55-point improvement in Perception task (i.e., Position, Color, Poster, Scene, Landmark, OCR) compared to Minigpt-v2. Although, there was an 8.21-point decrease in Cognition tasks, since PATCH was only fine-tuned on the POPE dataset, we believe our method could achieve better generalization through further training on other large-scale datasets.

---

### Official Review · Reviewer_MaRz · 2024-11-03

**Soundness:** 2
**Presentation:** 3
**Contribution:** 2
**Rating:** 6
**Confidence:** 4

**Summary:**

This paper addresses hallucinations in Large Vision-Language Models (LVLMs), proposing that they stem more from ineffective feature decoupling than a lack of visual input understanding. The authors introduce PATCH, a tuning method that employs adaptive virtual tokens to improve feature extraction from bounding boxes, effectively reducing hallucinations. PATCH claims that it achieves state-of-the-art results across multi-modal hallucination datasets, offering new insights into the architectural causes of hallucinations in LVLMs.

**Strengths:**

1. **Parameter Efficiency**: PATCH is designed to be parameter-efficient, reducing the need for extensive model adjustments while achieving high performance.
2. **Performance on Hallucination Datasets**: The proposed PATCH method achieves state-of-the-art results across multiple multi-modal hallucination datasets, demonstrating its effectiveness and robustness.
3. **Clarity and Simplicity**: The methodology is straightforward, and the paper is well-structured, making it easy to understand and follow.

**Weaknesses:**

1. **Lack of Novelty in Methodology**: The approach relies partially on using a pretrained object detection model for feature recognition and fine-tuning the vision-language model (VLM), which is a common strategy in previous works such as GLaMM, LISA, and Groundhog. This makes the method less innovative in this aspect.

2. **Insufficient Ablation Study on Virtual Tokens**: The paper introduces virtual tokens as part of the PATCH tuning strategy but lacks comprehensive ablation studies to validate their effectiveness, limiting insight into how much these tokens specifically contribute to mitigating hallucinations.

3. **Limited Plug-and-Play Functionality**: Although advertised as plug-and-play, the method requires fine-tuning for each personalized model, which contradicts its plug-and-play claim and may increase implementation complexity.

4. **Lack of Baseline Comparisons**: The study does not adequately compare its approach against similar methods. Baseline plug-and-play strategies, such as visual prompt-based methods (e.g., Set-of-Mark and so on), as well as fine-tuning methods like GLaMM, Lisa, and Groundhog, are absent, making it difficult to assess the actual improvements PATCH offers.

**Questions:**

1. **Novelty**:  How does PATCH specifically improve over zero-shot visual prompts methods like Set-of-Mark and  fine-tuning methods like GLaMM, Lisa, and Groundhog?

2. **Virtual Tokens**: Is there an ablation study showing the effectiveness of virtual tokens?

3. **Plug-and-Play Claim**: Since fine-tuning is needed, how is PATCH truly plug-and-play?

4. **Baseline Comparisons**: Would including comparisons with similar methods, like visual prompts mehtods and other fine-tuning models, provide further context for PATCH’s effectiveness?

---

> ### Author Response · Authors · 2024-11-19
> **Response to Reviewer MaRz**
>
> Thanks for your positive feedback and constructive comments. We would like to address your concerns in weaknesses one by one as follows.
>
> **W1**: Lack of Novelty in Methodology.
>
> **Response**: Thanks for your comments. We acknowledge that "utilizing a pretrained model for feature extraction and fine-tuning the vision-language model" is indeed a widely adopted paradigm for improving LVLM capabilities. However, the primary contribution of our research lies in identifying the specific module responsible for hallucination generation in LVLMs, rather than proposing a general approach for enhancing the LVLM performance.
>
> Our preliminary experiments reveal that hallucinations primarily arise in the modal alignment module. To address this issue, we propose the PATCH strategy to enhance the alignment between visual and textual features. PATCH is a plug-and-play method that can be dynamically adjusted based on the requirements of specific applications. Furthermore, we add more baselines you mentioned (as below), and the experimental results consistently demonstrate superior performance of our method.
>
> |  Method  | Accuracy | Precision | Recall | F1 |
> |:------:|:----:|:------:|:------:|:------:|
> | SoK[1] | 75.13 | 90.19 | 56.40 | 69.40 |
> | Groundhog[2] | 86.33 | 85.93 | 86.63 | 86.28 |
> | Qwen2-VL-7B[3] | 86.83 | 88.02 | 85.27 | 86.62 |
> | LLaVA OneVision-7B[4] | 87.13 | 90.25 | 83.27 | 86.62 |
> | GLaMM[5] | 87.30 | 85.03 | 90.53 | 87.70 |
> | **PATCH** | **90.03** | **91.39** | **88.40** | **89.87** |
>
> **W2**: Insufficient Ablation Study on Virtual Tokens.
>
> **Response**: We are sorry for the unclear organization of our ablation study. We have conducted a thorough ablation study in our paper. For clarity, we explain the results of all ablation experiments here, and all of them are based on MiniGPT-v2.
> **(1) origin**: We perform the zero-shot inference with the prompt: \<Img>[Image]\</Img>[vqa][Question].
> **(2) origin + objs**: We perform the zero-shot inference with the prompt: \<Img>[Image]\</Img>Objects:[Object][vqa][Question], where the object category information is included at the [Object] position, formatted as "category".
> **(3) origin + objs & bboxes**: We perform the zero-shot inference with the prompt: \<Img>[Image]\</Img>Objects:[Object][vqa][Question], where the object category and position information is included at the [Object] position, formatted as "category\{<x1><x2><x3><x4>\}".
> **(4) origin(sft) + objs & bboxes**: We fine-tune on MiniGPT-v2 using LoRA with the prompt: \<Img>[Image]\</Img>Objects:[Object][vqa][Question], where the object-related information is included at the [Object] position, formatted as "category\{<x1><x2><x3><x4>\}".
> **(5) PATCH**: Our method using 20 virtual tokens based on MiniGPT-v2.
> **(6) PATCH - bboxes:** Our method using on 20 virtual tokens based on MiniGPT-v2 with only the object category information.
> **(7) PATCH - bboxes & objs:** Our method using on 20 virtual tokens based on MiniGPT-v2 without any object-related information.
>
> Among these ablation studies, (1), (2), and (3) are proposed to evaluate the impact of different formats of detection results on MiniGPT-v2's inference. (4) is proposed to compare the effects between two different fine-tuning methods: our virtual tokens and LoRA. (5), (6), and (7) are proposed to evaluate the impact of different formats of detection results on PATCH.
>
> |  Method  | Accuracy | Precision | Recall | F1 |
> |:------|:----:|:------:|:------:|:------:|
> | (1) origin  | 83.37 | 88.00 | 77.27 | 82.29 |
> | (2) origin + objs | 88.80 | 88.19 | 89.60 | 88.89 |
> | (3) origin + objs & bboxes | 89.33 | 89.81 | 88.73 | 89.27 |
> | (4) origin(sft) + objs & bboxes | 87.97 | 88.45 | 87.33 | 87.89 |
> | (5) **PATCH** | **90.03** | **91.39** | **88.40** | **89.87** |
> | (6) PATCH - bboxes  | 88.27 | 87.37 | 89.47 | 88.41 |
> | (7) PATCH - bboxes & objs | 82.60 | 84.93 | 79.27 | 82.00 |
>
> **Reference**
> [1] Yang, Jianwei, et al. "Set-of-mark prompting unleashes extraordinary visual grounding in gpt-4v." arXiv preprint arXiv:2310.11441 (2023).
> [2] Zhang Y, Ma Z, Gao X, et al. Groundhog: Grounding large language models to holistic segmentation[C]//Proceedings of the IEEE/CVF conference on computer vision and pattern recognition. 2024: 14227-14238.
> [3] Wang, Peng, et al. "Qwen2-vl: Enhancing vision-language model's perception of the world at any resolution." arXiv preprint arXiv:2409.12191 (2024).
> [4] Li, Bo, et al. "Llava-onevision: Easy visual task transfer." arXiv preprint arXiv:2408.03326 (2024).
> [5] Rasheed, Hanoona, et al. "Glamm: Pixel grounding large multimodal model." Proceedings of the IEEE/CVF Conference on Computer Vision and Pattern Recognition. 2024.

---

> ### Author Response · Authors · 2024-11-19
> **Response to Reviewer MaRz**
>
> **W3**: Limited Plug-and-Play Functionality.
>
> **Response**: We understand your concern. Due to its lightweight design, PATCH can be used in a plug-and-play manner. For tasks prone to hallucinations, such as VQA, virtual tokens can be added to significantly enhance the performance. Especially for tasks highly related to object attributes, we conducted relevant experiments on VQAv2, and the results are presented as follows. In contrast, in scenarios such as content understanding or relational reasoning, virtual tokens can be removed to maintain the original capabilities of the LVLM. This is what we call "plug-and-play", and such flexibility ensures that PATCH remains practical and adaptable for LVLMs already deployed in real-world applications.
>
> | Question Categories | origin Accuracy | PATCH Accuracy|
> |:------|:----:|:------:|
> | what is this | 70.91 | **75.40** |
> | what are | 67.51 | **69.28** |
> | what sport is | 93.28 | **95.47** |
> | what animal is | 85.52 | **87.20** |
> | what is the name | 30.77 | **31.23** |
>
> **W4**: Lack of Baseline Comparisons.
>
> **Response**: Thank you for your valuable feedback. According to your suggestions, we add new comparisons with similar methods, including baseline plug-and-play strategies, such as visual prompt-based methods (i.g., Set-of-Mark, GAVIE, Osprey) and fine-tuning methods (i.e. Groundhog, VCD,Chat-uniVi, GoundingGPT). The results are shown below.
> |  Method  | Accuracy | Precision | Recall | F1 |
> |:------|:----:|:------:|:------:|:------:|
> | SoK[1] | 75.13 | 90.19 | 56.40 | 69.40 |
> | Groundhog[2] | 86.33 | 85.93 | 86.63 | 86.28 |
> | GAVIE[3] | 62.30 | 57.46 | 94.66 | 71.51 |
> | LLaVA1.5+VCD[4] | 80.88 | 79.45 | 83.29 | 81.33 |
> | Chat-UniVi-7B[5] | 64.97 | - | - | 71.54|
> | Chat-UniVi-7B-v1.5[5] | 83.23 | - | - | 82.31 |
> | GoundingGPT[6] | 86.17 | - | - | 85.50 |
> | Osprey[7] | 85.33 | 85.43 | 85.20 | 85.31 |
> | GLaMM[8] | 87.30 | 85.03 | 90.53 | 87.70 |
> | **PATCH** | **90.03** | **91.39** | **88.40** | **89.87** |
>
> Compared to these methods, it is clear to see that our approach still achieves the state-of-art results, demonstrating its superior performance and advantages in addressing the object hallucination issue in LVLMs.
>
>
> **Reference**
> [1] Yang, Jianwei, et al. "Set-of-mark prompting unleashes extraordinary visual grounding in gpt-4v." arXiv preprint arXiv:2310.11441 (2023).
> [2] Zhang Y, Ma Z, Gao X, et al. Groundhog: Grounding large language models to holistic segmentation[C]//Proceedings of the IEEE/CVF conference on computer vision and pattern recognition. 2024: 14227-14238.
> [3] Liu, Fuxiao, et al. "Mitigating hallucination in large multi-modal models via robust instruction tuning." The Twelfth International Conference on Learning Representations. 2023.
> [4] Leng, Sicong, et al. "Mitigating object hallucinations in large vision-language models through visual contrastive decoding." Proceedings of the IEEE/CVF Conference on Computer Vision and Pattern Recognition. 2024.
> [5] Jin, Peng, et al. "Chat-univi: Unified visual representation empowers large language models with image and video understanding." Proceedings of the IEEE/CVF Conference on Computer Vision and Pattern Recognition. 2024.
> [6] Li, Zhaowei, et al. "Groundinggpt: Language enhanced multi-modal grounding model." Proceedings of the 62nd Annual Meeting of the Association for Computational Linguistics (Volume 1: Long Papers). 2024.
> [7] Yuan, Yuqian, et al. "Osprey: Pixel understanding with visual instruction tuning." Proceedings of the IEEE/CVF Conference on Computer Vision and Pattern Recognition. 2024.
> [8] Rasheed, Hanoona, et al. "Glamm: Pixel grounding large multimodal model." Proceedings of the IEEE/CVF Conference on Computer Vision and Pattern Recognition. 2024.

---

> ### Author Response · Authors · 2024-11-22
> **To confirm whether our responses has addressed the concerns and thank you for your efforts**
>
> Dear Reviewer,
>
> We sincerely appreciate your time and effort in reviewing our paper and offering valuable suggestions. Since there will be no second stage of author-reviewer discussions, and the discussion phase is drawing to a close, we would to confirm whether our responses have effectively addressed your concerns a few days ago. And we hope they have adequately addressed your issues. If you require further clarification or have additional concerns, please don’t hesitate to contact us. We are more than willing to continue our communication with you.
>
> Best regards.

---

> > ### Comment · Reviewer_MaRz · 2024-11-22
> >
> > Thanks for the reply. Most of my concerns have been addressed. I will improve my score.

---

> > > ### Author Response · Authors · 2024-11-23
> > > **Thanks for the reply**
> > >
> > > We sincerely thank you for your recognition of this paper and your valuable suggestions. We’re glad we could address your concerns.

---

### Meta-Review · Area_Chair_hvAh · 2024-12-20

**Metareview:**

This work analyses the reason for LVLM object hallucination, and find that the visual feature decoupling rather than extraction is the main cause. The authors thus designed PATCH which is a plug-and-play prompt-tuning-based approach. The results demonstrated the efficacy of such a method. The paper got diverse recommendations from the four reviewers (2 acceptance and 2 rejection). The major concerns include: (1) The approach mainly involves parameter-efficient tuning, and it does not bring obvious novelty over existing tuning methods; (2) the validity of the experimental designs is not convincing; and (3) there are weaknesses in terms of the used baselines and datasets in experiments. Considering these aspects, AC recommends rejection.

**Additional Comments On Reviewer Discussion:**

After rebuttal, reviewers still have concern about the novelty of this work, and the limitations and weaknesses in the experiments.

---

### Decision · Program_Chairs · 2025-01-22

Reject